

# Breeze effects at a large artificial lake: summer case study

Maksim Iakunin[1], Rui Salgado[1], and Miguel Potes[1]

[1]Department of Physics, ICT, Institute of Earth Sciences, University of Évora, 7000 Évora, Portugal

*Correspondence to:* Maksim Iakunin (miakunin@uevora.pt)

**Abstract.** Natural lakes and big artificial reservoirs could affect the weather regime of surrounding areas but usually it is difficult to track all aspects of this impact and evaluate its magnitude. Alqueva reservoir, the largest artificial lakes in Western Europe located on the South-East of Portugal, was filled in 2004. This makes it a large laboratory and allows to study the changes in hydrological and geological structures and how they affect the weather in the region. This paper is focused on
a case study of the 3 days period of 22-24 July 2014. In order to quantify the breeze effects induced by Alqueva reservoir two simulations with the mesoscale atmospheric model Meso-NH coupled to FLake freshwater lake scheme has been done. The principal difference of this two simulations is in the presence of the reservoir in the input surface data. Comparing two simulations datasets: with and without reservoir, net results of the lake impact were obtained. Magnitude of the impact on the air temperature, relative humidity, and other atmospheric parameters is shown. Clear effect of a lake breeze (5-7 m/s) can be
observed during the daytime on the distances up to 6 km away from the shores and up to 300 m over the lake surface. Breeze system starts to form at 9:00 UTC and dissipates at 18:00-19:00 UTC with the arrival of major Atlantic breeze system. It induces specific air circulation that captures the dry air from the upper atmosphere (2-2.5 km) which follows the downstream and redistributes over the lake. It is also shown that the although the impact can be relatively intensive, its area is limited by several kilometers away from the lake borders.

## 1  Introduction

Human activity, such as urbanization, deforestation or water reservoirs building, changes surface properties (vegetation cover, emissivity, albedo) which determine surface energy fluxes (Cotton and Pielke, 2007). As a consequence, changes in surface energy fluxes affect local weather and climate. Lakes and reservoirs contains about 0.35 % of global freshwater storage (Hartmann, 1994) and cover only 2% of continental surface area (Segal et al., 1997). However, they play a huge soci-
etal role. Thermal circulations triggered by lake/land thermal contrast impact on dispersion of air pollution and lake catchment transport (Lee et al., 2014). Big lakes being a significant source of atmospheric moisture can intensify storm formation (Samuelsson et al., 2006; Zhao et al., 2012). Lakes and reservoirs can be characterized by increased thermal inertia and heat capacity, small albedo and roughness length compared to vegetated land surfaces (Bonan, 1995). They can affect meteorological conditions and atmospheric processes at meso and synoptic scales (Pielke, 1974; Bates et al., 1993; Pielke, 2013).
Normally, surface moisture is increased while daily air temperature near the surface is decreased in lake shore areas. During the warm summer period relatively colder lake surface balances the atmosphere above, forcing a reduction of clouds and pre-



cipitation. Formation of the local high pressure areas over the lake surface in summer season supports atmospheric circulation, which can be observed as a lake breeze (Bates et al., 1993). In autumn and winter it has the opposite effect due to the warmer air above lake surface: increase of evaporation and cloud formation (Ekhtiari et al., 2017). These lake effects on the regional climate regime find confirmation in previous studies, e.g. Elqui Valley reservoir in Chile (Bischoff-Gauß et al., 2006) and the

great African lakes (Thiery et al., 2014).

Despite the fact that the theoretical aspects of formation of the lake breezes are clear, in practice, they remain not well documented. Difficulties in studies of lake breeze are due to the diversity and complexity of lake shapes and surrounding landscapes, and the using of coarse spatial resolution observations data (Segal et al., 1997).

Lake breezes are mainly determined by geophysical variables and weather conditions. Formation and intensity of the breeze

depend on the set of parameters such as large scale winds, sensible heat flux, geometry of the lake, terrain types of the surrounding area, and others (Segal et al., 1997; Drobinski and Dubos, 2009; Crosman and Horel, 2012).

In this work the focus is on the study of the lake of Alqueva and its impact on atmospheric parameters of the surrounding area. This large artificial reservoir has been filled in 2004 which makes it a big natural laboratory for studying physical, chemical, and biological effects. Few studies about the influence of Alqueva on atmosphere and climate were published. A

first report, in Portuguese, was published even before the construction of the dam by Miranda et al. (1995), as part of the environmental impact study of the reservoir. They concluded, on the basis of numerical simulations performed with the NH3D model Miranda and James (1992) that the climate impact of the the multi-purpose Alqueva project should be essentially due to the projected irrigation area and pay little attention to the reservoir itself as at the time it was not possible to perform high resolution simulations. The studies were continued and improved by Salgado (2006) in his PhD thesis, also in Portuguese, in

which a first attempt to quantify the direct effect of the reservoir on the local climate, in particular on winter fog, were done. Using the Meso-NH model, the author concluded that the introduction of the reservoir should increase slightly the winter fog in the neighborhood, but decrease over the filled area. Later, Policarpo et al. (2017) used observations for two periods (before and after Alqueva) and also Meso-NH simulations, showing a slight increase in the average number of days with fog during the winter (DJF), of about 4 days per winter after 2003 in a downwind site and reinforcing previous findings.

On the other hand, data collected in and above the Alqueva reservoir allowed the characterization of energy and mass transfers between the water and the air (Salgado and Le Moigne, 2010; Potes et al., 2012) and were used to calibrate the FLake model (Mironov, 2008) and to validate its integration in the SURFEX platform of surface models Masson et al. (2013) used among other atmospheric models by Meso-NH.

Using a mesoscale atmospheric model as Meso-NH allows to gain the results with sufficient horizontal resolution (up to 250

m) to track the local effects of air temperature changes and small-scale winds on the background of large-scale atmospheric motions. Simulation has been done for the Intensive Observation Period (IOP) of an in-situ measurement field campaign on the lake area (ALEX2014 — ALqueva hydro-meteorological EXperiment, http://www.alex2014.cge.uevora.pt/), so it allowed to validate the acquired results with different datasets.

The article outline is the following. Section 2 provides a brief description of the Alqueva reservoir, the object of current

study. Section 3 reveals information about ALEX2014 experiment and dataset used in this paper: meteorological stations,



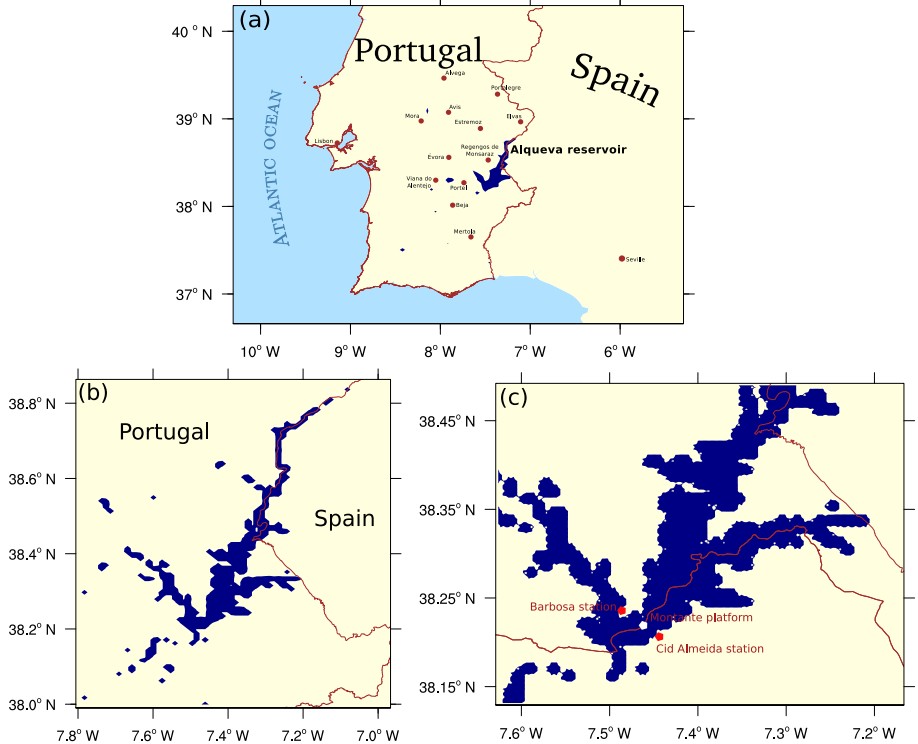

**Figure 1.** Maps of the nesting domains used in the simulations: **(a)** 4 km horizontal resolution, $100{\times}108$ pixels, with location of 12 IPMA synoptic stations used for validation process **(b)** 1 km horizontal resolution, $96{\times}72$ pixels, **(c)** 250 m horizontal resolution, $160{\times}160$ pixels, with ALEX2014 land stations and floating platform Montante location.

observations and measurements. Section 4 contains a brief description of the numerical models: Meso-NH and FLake, used in this work. Sections 5 and 6 are dedicated to the case study on 22-24 July 2014: validation of simulation results using in-situ measurements and the studies of the lake effects respectively, with an illustration and discussion of the magnitude of the impact and intensity of a lake breeze. Section 7 summarizes the results and conclusions.

## 5  2   Object of study

Alqueva reservoir established in 2002 is an artificial lake located in the South-East part of Portugal. It spreads along 83 km over the Guadiana river valley covering the total area of 250 km$^2$ with the total capacity if 4.15 km$^3$ which makes it the largest artificial lake in Western Europe (Fig. 1 (a)). The maximum and average depths of the reservoir are 92.0 m and 16.6 m respectively.

10   Alqueva reservoir is mainly used to provide water supply, irrigation, and hydroelectric power. Surrounding region is known for the irregularity of its hydrological resources, with the long periods of drought that could last for more than one consecutive year (Silva et al., 2014). This region is characterised as Csa according to the Köppen climate classification (Mediterranean



climate with dry and hot summers), with a small area within the mid-latitude steppe (BSk) category. During summer, the maximum air temperature ranges between 31 and 35°C on average (July and August), often reaching values close to 40°C, or even higher. The incident solar radiation on the surface level is of the highest in Europe, with mean daily values (integrated over 24 hours) of about $300\,\mathrm{Wm^{-2}}$ and the daily maximum of July often can reach $1000\,\mathrm{Wm^{-2}}$. Rainfall periods are seasonal and last from October to April, an average annual precipitation registered in the city of Beja ($40\,\mathrm{km}$ from Alqueva reservoir) over 1981-2010 is $558\,\mathrm{mm}$ (www.ipma.pt).

Two major factors determine synoptic circulations over the region during the summer period: the shape and location of the Azores anticyclone, and the frequent establishment of a low-pressure system over the Iberian Peninsula inside it, induced by the the land-ocean thermal contrasts. The sea breeze system controls the transport of the maritime air masses from the Atlantic coast of the peninsula to its internal areas on distances more than $100\,\mathrm{km}$ reaching the Alqueva region in the late afternoon. This phenomenon is known as the Iberian thermal low (Hoinka and Castro, 2003) and characterized by a westward change of the wind direction (prevailing wind directions are from the North-West quadrant). As a result, this effect is observed in the local increase in wind intensity and its rotation (Salgado et al., 2015).

## 3 Dataset

The main dataset for this work has been obtained during the ALEX2014 campaign — a multidisciplinary observational experiment at the Alqueva reservoir which has last from June to October 2014. The aim of this project included a wide set of measurements of chemical, physical, and biological parameters in the water, air columns, and over the water-atmosphere interface. To reach this goal the project operated the following facilities:

- 7 sites with meteorological measurements: 2 Platforms (Montante and Mourão); permanent weather station in the island (Alquilha), 2 dedicated weather stations (Barbosa and Cid Almeida), two compact weather stations in the Solar Park and Amieira;

- 4 floating platforms where water quality and biological sampling were done: Montante, Mourão, Captação and Alcarrache;

- 3 stations of Instituto de Ciências da Terra (ICT), located in Mitra, Portel, and at the University of Évora;

- 2 Air quality mobile units: Amieira and at Solar Park;

- 3 Atmospheric Electricity stations: Amieira, Solar Park and Beja.

Also data from 42 IPMA (Portuguese Institue of Sea and Atmosphere) meteorological stations located over all nearby regions were integrated into ALEX database. They provided basic set of parameters, e.g. air temperature, relative humidity, pressure, horizontal wind speed.

Two land stations (Barbosa and Cid Almeida) were located on the opposite shores and the floating platform Montante in the middle (Fig. 1 (c)). This locaton allowed to monitor the lake effects in real time during the observations. Land stations collected





the following data with 1-minute time resolution: horizontal wind speed, relative humidity, air temperature, downwelling short-wave radiation, and precipitation. The floating platform Alqueva Montante was the principal experimental site inside the reservoir. The following equipment was installed there and collected data since 2 June 2014 until the end of the campaign:

- Irgason eddy-covariance system which provides data for: pressure, temperature, water vapor and carbon dioxide concentrations, 3D wind components, momentum flux, sensible and latent heat fluxes, carbon dioxide flux, evaporation;

- one albedometer and one pirradiometer in order to measure upwelling and downwelling shortwave and total radiative fluxes;

- 9 thermistors to measure water temperature profile.

The most intensive observational part of the ALEX project has last 3 days (22-24 July) and included launches of meteorological balloons every 3 hours. For that TOTEX meteorological balloons (600 g) were used. In total, 18 radiosondes have been launched: 2 from the boat over the lake and 16 from the land. Atmospheric profiles of air temperature, relative humidity, wind, and pressure were obtained with the use of Vaisala Radiosondes model RS92-SGP. This period, 22-24 July, was chosen for a case study in the this work, because it was a period with mostly atmospheric stability and anticyclonic conditions.

Data collected during the ALEX2014 field campaign have already been used to study: lake-atmosphere interactions, including the heat and mass ($H_2O$ and $CO_2$) fluxes in the interface water-air (Potes et al., 2017); the effects of inland water bodies on the atmospheric electrical field (Lopes et al., 2016); and the evolution of the vertical electrical charge profiles and its relation with the boundary layer transport of moisture, momentum and particulate matter (Nicoll et al., 2018).

## 4 Simulation setup

### 4.1 Meso-NH atmospheric model

For the study of the breeze effects of the Alqueva reservoir the Meso-NH model (Lac et al., 2018) was used.

Meso-NH is a non-hydrostatic mesoscale atmospheric research model. It has a complete set of physical parameterizations, which are particularly advanced for the representation of clouds and precipitation, incorporates a non-hydrostatic system of equations, for dealing with scales ranging from large (synoptic) to small (large eddy) scales while calculating budgets. Meso-NH is coupled with SURFEX (Masson et al., 2013) platforms of models for the representation of surface-atmosphere interactions by considering different surface types (vegetation, city, ocean, lake), and allows a multi-scale approach through a grid-nesting technique (Stein et al., 2000).

Three nesting levels were used in Meso-NH runs: $400 \times 432\,\mathrm{km}^2$ domain with 4 km spatial resolution to take into account the large scale circulations, namely the influence of the sea breeze (Fig. 1 (a)), $96 \times 72\,\mathrm{km}^2$ domain with 1 km spatial resolution centered at the Alqueva reservoir (Fig. 1 (b)), and a small $40 \times 40\,\mathrm{km}^2$ domain with 250 m spatial resolution to track the minor effects of the lake (Fig. 1 (c)). Hereinafter we denote this three domains A, B, and C correspondingly.



**Table 1.** Summary of the Meso-NH physical schemes used in the simulations.

| Schemes and parameters | 4-km domain | 1-km domain | 250-m domain |
| --- | --- | --- | --- |
| Deep convction | KAFR | NONE | NONE |
| Shallow convection | EDKF | EDKF | NONE |
| Turbulence | BL89 1 dimension | DEAR 3 dimensions | DEAR 3 dimensions |
| Radiation transfer | ECMW | ECMW | ECMW |
| Advection | WENO | WENO | WENO |
| Clouds | ICE3 | ICE3 | ICE3 |
| Timestep | 20 s | 5 s | 1 s |

For the surface and orography ECOCLIMAP II (Faroux et al., 2013) and SRTM (Jarvis et al., 2008) databases were used, respectively, both updated with the presence of Alqueva reservoir by Policarpo et al. (2017). All model domains had 68 vertical levels starting with 20 m and up to 22 km at the top, including 36 levels for the lower atmospheric level. The model configuration included the turbulent scheme based on a one-dimensional 1.5 closure (Bougeault and Lacarrere, 1989). Mixed

microphysical scheme for stratiform clouds and explicit precipitation (Cohard and Pinty, 2000; Cuxart et al., 2000) which distinguish 6 classes of hydrometeors (water vapor, cloud water droplets, liquid water, ice, snow, and graupel). Longwave and shortwave radiative transfer equations are solved for independent air columns (Fouquart and Bonnel, 1980; Morcrette, 1991). Atmosphere-surface flux exchange controlled by physical parametrisations: the surface soil and vegetation are described by the Interface Soil Biosphere Atmosphere (ISBA) model (Noilhan and Mahfouf, 1996); the town energy balance was handled

according to Masson (2000). Basic parameters for each model domain are shown in the Table 1. 4-km horizontal resolution in the first domain is coarse enough to use deep and shallow convection schemes in simulation. 1-km resolution of the second domain already required deep convection to be resolved explicitly. 250-m resolution is fine so both schemes can not be applicable. Used schemes (see Table are the following: KAFR (Kain and Fritsch, 1990; Bechtold et al., 2001), EDKF (Pergaud et al., 2009), WENO (Lunet et al., 2017), ICE3 (Pinty and Jabouille, 1998).

European Centre for Medium-Range Weather Forecast (ECMWF) operational analyses data files were used for Meso-NH initialization and lateral boundary forcing, updated every six hours.

To track the direct impact of the reservoir on the weather conditions, a set of two numerical simulation were performed: the one with the surface input files updated to Alqueva reservoir presence (ECOCLIMAP database version updated by Policarpo et al. (2017)) and another — with the previous version of this database where the reservoir does not exist yet. In order to distinguish

these simulations hereinafter we denote them LAKE1 and LAKE0, correspondingly. Both simulations have covered the case





study period, 22-24 July 2014, with 1 hour output. To reproduce atmospheric conditions more realistic the simulation included previous 24 hours (21 July), so, overall the model covered 96 hours. The differences between these two simulations were then computed, with the aim of evaluating the direct influence of the presence of the lake on the environment.

### 4.2 FLake scheme

In order to represent the presence and evolution of the lake more realistically, freshwater lake model FLake (Mironov, 2008) were used. FLake is a bulk-type model capable to predict the evolution of the lake water temperature at different depth on time scales from a few hours to many years. For the unfrozen lake it uses two-layer approach: upper mixing layer with the constant water temperature from the surface and the thermocline level beneath it where the temperature decreases with depth. Parametrization of the thermocline layer is based on the concept of self-similarity assuming that such approach could be applied
to all natural and artificial freshwater lakes.

     The following parameters are required to run the FLake model: four initial parameters of the lake temperature structure, six atmospheric parameters for each calculation timestep, and two constants — lake depth and the attenuation coefficient of light into water. This coefficient represents the penetration of the solar radiation in the water body. In this work attenuation coefficient was set to 0.85 corresponding the in-situ measurements.

Initial parameters are: water temperature at the bottom, temperature and depth of the mixing layer, and shape factor $C_f$ — specific parameter that describes the shape of the the thermocline curve. In the parametrization proposed by Mironov (2008) for normalized temperature profile and depth it varies from 0.5 to 0.8. The shape factor $C_f$, temperature of the water mixing layer and its depth were determined using fitting technique of real water temperature profile data from Montante platform (Fig. 2). FLake model is very sensitive to initial parameters, and this fitting technique is based on assumption that the bottom
temperature is fixed by the value of the lowermost sensor. Thereby, the other three parameters could vary within some range until the best set of them is found. The set of parameters for this case study is: $C_f = 0.8$, mixed layer temperature is 23.8 °C, and depth is 3.4 m.

     In view of the fact that the observed daytime temperature profiles showed strong skin effects (higher temperatures of the upper (up to 10 cm) water layer) and could not be correctly fitted, midnight profile was used as an initial one and the simulation
started at midnight, 21 July 2014.

     Required atmospheric parameters: horizontal wind speed, air temperature, special humidity, longwave and shortwave downwelling radiation, and atmospheric pressure were taken directly from Meso-NH simulation since FLake was implemented in SURFEX model (Salgado and Le Moigne, 2010).

     Typically, the depth of the artificial lakes characterized by strong spatial variability, because the lake bottom of these lakes
used to be a valley. In case of Alqueva, when completely filled, the mean depth is of about 17 m (http://www.edia.pt/). On the other hand, the local depth at Montante platform can reach 70 m. As an 1D bulk model, FLake has only one depth value which should be seen as an effective depth and is not easy to assess. Moreover, FLake scheme is not capable to represent deep lakes, the scheme works well for depths from 20 to 50 m with the sediments routine switched off. After a series of a sensitivity tests of short-term (2-4 days) and long-term (2-4 months) simulation it was found that the best simulations results can be obtained





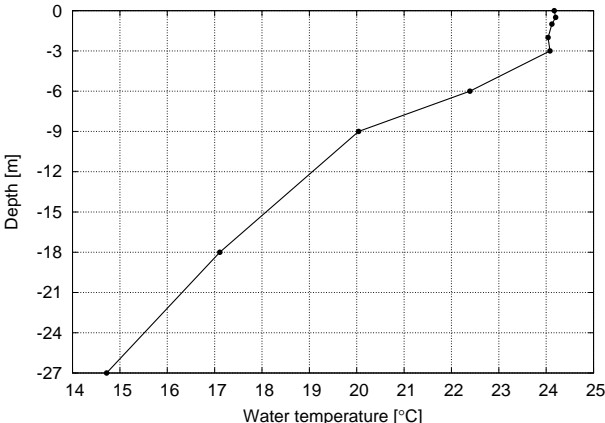

**Figure 2.** Water temperature profile on July 21, 00:00 at Montante platform.

with the bottom depth value of 20-30 m. Thus, since the last profile sensor was at the depth of 27 m, this value was chosen for the effective lake depth in this work.

## 5 Validation

The simulation LAKE1 results were validated against radiosondes data (vertical profiles) and meteorological stations data:
ALEX and IPMA synoptic stations and floating platforms. In this process all three domains were used. Size of the domain A was enough to consider 12 meteorological stations in the region, domain B was used to track the radiosondes trajectory, and domain C data was used for the validation against stations at the lake shores and Montante floating platform. The parameters to compare are: air temperature, relative humidity, wind speed, sensible and latent heat fluxes.

### 5.1 Comparison with radiosondes data

The ALEX 2014 IOP of 22 – 24 of July 2014 took place at the Alqueva reservoir. They included balloon launches every 3 hours, and overall 18 launches have been done. Balloons radiosonde data provided air temperature, humidity, pressure, and wind speed.

Due to the fact that the trajectory of the balloons was not vertical but resembled a spiral and balloons flew several kilometers away from the launching point it was decided to make a trajectory profile comparison. Each balloon had a GPS-tracker to
15 register its coordinates every 2 seconds, which was used to build a corresponding trajectory inside the simulation domain. Radiosondes have reached the altitude of $\approx 35$ km, but since the upper layer of the model is limited to 22 km, the profiles were built up to this altitude. For sondes it took about 2.5 hours to reach 22 km height so to build a corresponding profile three consecutive timestep arrays of the model data were used. This comparison is done in 1-km horizontal resolution domain.

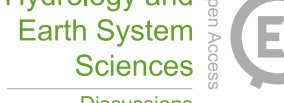



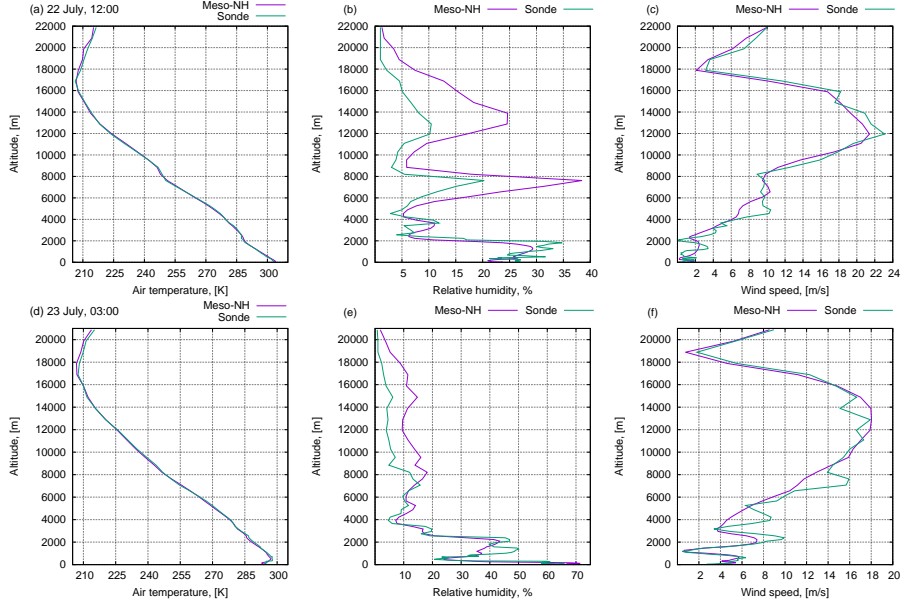

**Figure 3.** Examples of vertical profiles for July 22, 12:00 and July 23, 3:00 of air temperature (a, d), relative humidity (b, e), and wind speed (c, f).

Figures 3 (a,b,c) represent examples of the daytime profiles for air temperature, relative humidity, and horizontal wind speed. Examples of corresponding night profiles can be found on figures 3 (d,e,f). All night profiles demonstrate slightly better accordance with model results because atmosphere is more stable during the nighttime.

Figure 4 represents the same profiles for the lower atmospheric level (3000 m altitude). Simulation results are in good
accordance with measured values under the radiosonde accurace ($\pm 0.5°$ C, $\pm 5\%$ relative humidity, and $\pm 0.15$ m/s wind speed with 2-sigma confidence level (95.5 %)).

The principal features of the curve profiles as well as the dynamics are well represented by the model. On air temperature curves (and, to a lesser extent, on the relative humidity and wind speed curves), one can observe a characteristic fracture at 2-2.5 km altitude during daytime, which denotes the top of the boundary layer. Overall, Meso-NH better represents air
temperature in the layer from 2.5 km to 10-12 km, while the worst values are in the lower lever near the surface during the period of 06:00 - 12:00 UTC (presented in suplementary material).

Patterns of relative humidity and wind speed are good as observed and modelled curves look similar, nevertheless simulations tend to be more conservative and their values do not change so quick. For example, low level jets can be found on night profiles at the edge of the boundary level (Fig. 4 (f)). These jets are represented by the model as well, but their magnitude are slightly
weaker than the observed values. All 18 profiles can be found in the suplementary material. Statistical results for them are following. Temperature average bias is -0.13 K, RMSE 1.49 K, and correlation coefficient is 0.99. Humidity average bias is 0.59% RMSE of 11.26 % and 0.87 correlation coefficient. For wind speed these values are: 0.05 m/s average bias, RMSE





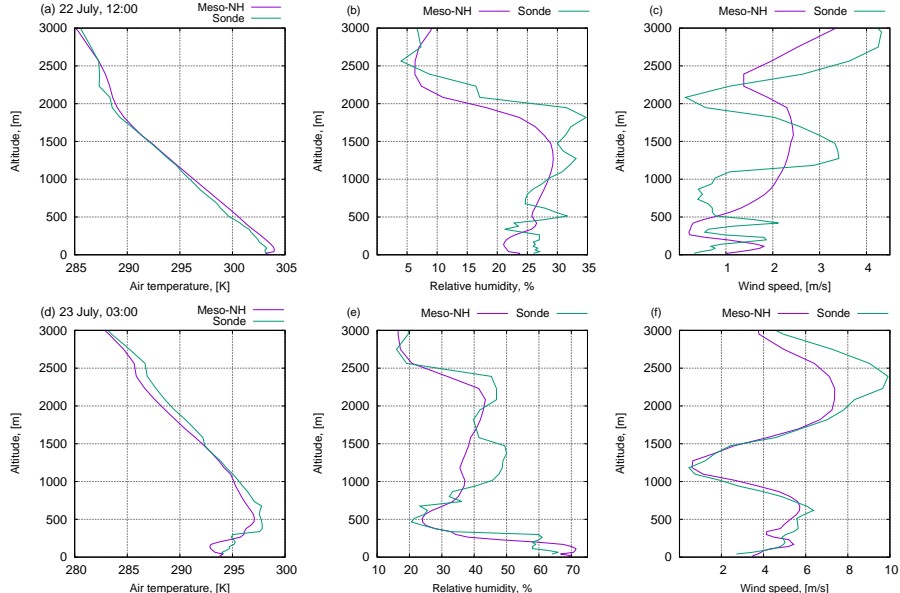

**Figure 4.** Lower atmospheric level profiles for air temperature, relative humidity, and wind speed.

is 2.07 m/s, and correlation coefficient is 0.90. All these values show that the simulation is in a good accordance with the observations, and in line with similar studies of Meso-NH validation against radiosondes data (e.g., Masciadri et al., 2013).

## 5.2 Comparison with IPMA stations data

We also validated the model data against 12 IPMA automatic meteorological stations. For this comparison data of 4-km

domain were used. Geographical positions of the stations can be found on Fig. 1 (a). Scatter plots of air temperature, relative humidity, and wind speed shown on Fig. 5. Not all stations provided the same set of parameters. These scatter plots represent the intercomparison of the model data (X axis) and the measured values (Y axis) over the case study period. The model tends to overestimate lower values of air temperature (14-24 °C) and slightly underestimate higher values (>30 °C), visible in Fig. 5 (a).

For relative humidity the model shows lower values within the range from 40 to 100 %. Wind speed is overestimated by the model as the output is set to 10 m above the surface and measurements are at 2 m.

Statistical parameters (biases, mean absolute errors, root mean square, and correlation coefficients) of this comparison are shown in Table 2. Comparison of air temperature showed high correlation (correlation coefficient is higher that 0.91) with biases absolute values lesser than 1 degree. The worse result are observed in comparison against Portalegre data (see for

example the square points in relative humidity plot on Fig. 5 (b)): this station is located in small mountain area which makes the meteorological situation more difficult to predict. Validation of the wind speed shown the lowest correlation coefficient (0.65 in Portalegre, 0.82 average) due to its high variability over time. Overall, simulation results are in good agreement with



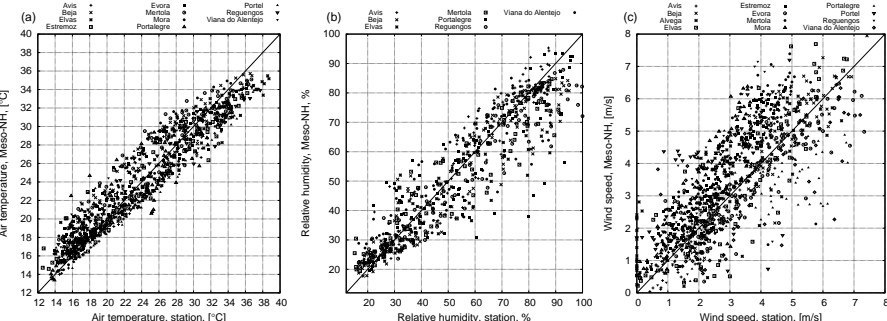

**Figure 5.** Scatter plots of the comparison with synoptic stations. Air temperature **(a)**, relative humidity **(b)**, and horizontal wind speed **(c)**.

**Table 2.** Statistics for the hourly values of the station validation.

| Stations: | | Alvega | Avis | Beja | Elvas | Estrem. | Évora | Mert. | Mora | Portal. | Portel | Reguen. | V. Alen. |
|---|---|---|---|---|---|---|---|---|---|---|---|---|---|
| Temp., | Bias: | — | -0.08 | 0.68 | -0.39 | 0.00 | 0.56 | 0.85 | 0.9 | -0.08 | -0.30 | -0.33 | 0.52 |
| K | MAE: | — | 1.49 | 1.60 | 1.76 | 1.65 | 1.60 | 1.71 | 1.54 | 1.82 | 1.91 | 1.44 | 1.82 |
| | RMS: | — | 1.84 | 1.96 | 2.18 | 2.02 | 1.96 | 2.20 | 1.93 | 2.38 | 2.27 | 1.82 | 2.13 |
| | Corr: | — | 0.95 | 0.96 | 0.96 | 0.96 | 0.96 | 0.95 | 0.96 | 0.91 | 0.94 | 0.97 | 0.96 |
| Rel. | Bias: | — | 0.53 | -2.98 | -3.42 | — | — | -1.29 | -4.19 | -2.79 | — | -4.10 | 1.80 |
| hum., | MAE: | — | 5.80 | 7.48 | 5.87 | — | — | 6.61 | 6.88 | 7.83 | — | 6.94 | 8.08 |
| % | RMS: | — | 7.41 | 9.49 | 8.61 | — | — | 8.49 | 8.43 | 11.91 | — | 9.11 | 9.80 |
| | Corr: | — | 0.93 | 0.93 | 0.93 | — | — | 0.94 | 0.94 | 0.86 | — | 0.95 | 0.90 |
| Wind | Bias: | 0.49 | 0.31 | 0.17 | 0.07 | 0.89 | -0.36 | 1.08 | 1.19 | -0.66 | 0.81 | 0.95 | -0.13 |
| speed, | MAE: | 0.91 | 0.60 | 0.60 | 0.74 | 0.97 | 0.80 | 1.25 | 1.30 | 1.05 | 1.16 | 1.13 | 0.93 |
| m/s | RMS: | 1.15 | 0.77 | 0.77 | 0.99 | 1.11 | 1.01 | 1.42 | 1.52 | 1.36 | 1.44 | 1.37 | 1.24 |
| | Corr: | 0.85 | 0.92 | 0.91 | 0.86 | 0.93 | 0.82 | 0.82 | 0.84 | 0.65 | 0.66 | 0.85 | 0.70 |

synoptic stations data. Other works represent the similar results of Meso-NH validation against data from meteostations (e.g., Lascaux et al., 2013, 2015).

### 5.3 Comparison with data from ALEX Lake platform and stations

In addition to the validation against the IPMA synoptic stations, comparisons were made with data obtained at ALEX2014
5 dedicated stations: Montante platform (38.2235° N, 7.4595° W) and two stations: Cid Almeida (38.2164° N, 7.4545° W) and Barbosa (38.2276° N, 7.4708° W) (Fig. 1 (c)). These coordinates were used to locate the stations on the 250-m output. Another criteria which was used for this was water fraction variable: land stations can not be found over the water, so the nearst land grid-point was used.





Figure 6 represents the evolution in time of air temperature and wind speed for these stations and the comparison to the corresponding simulated parameters. Meso-NH underestimation of air temperature in the afternoon time is opposite of wind speed overestimation at the same period. Values of wind speed show higher amplitude which may can be partlially explained by the same fact that model wind corresponds to the 10 m height while sensors at the station are at a height of 2 m. Statistical

values for this validation are the following. For Barbosa: air temperature maximum absolute error (MAE) is 3.6 K with RMSE of 1.37 K and correlation coefficient 0.98, and for wind speed: 4.4 m/s is maximum absolute bias, 2.13 m/s RMSE, and 0.67 correlation coefficient. For Cid Almeida: MAE 4.9 K, RMSE 1.57 K, correlation coefficient is 0.98, wind speed MAE is 5.9 m/s, RMSE is 1.56 m/s wid the correlation coefficient of 0.63. For Montante platform these values are following. Air temperature MAE is 3.2 K, RMSE 1.22 K, correlation is 0.98, wind speed MAE 4.95 m/s, RMSE 1.76 m/s, and correlation

coefficient is 0.63. Wind speed in the simulation input is given at 10 meters high while the stations are installed to measure it at 2 meters. Interpolation of the model results for 2 meters can reduce biases and improve the comparison. All maximums of the biases can be found on the peaks of the temperature or wind speed which is explained by the fact that the model is more conservative.

Dynamic of latent and sensible heat fluxes at Montante platform is shown on Fig. 7 (a, b). Overall patterns of these curves are

similar but simulated results are more smooth while observed values changes more quickly. Comparison between measurements and simulated results demonstrates that for latent heat RMSE is 57.34 $\mathrm{Wm}^{-2}$ with correlation coefficient of 0.47, and for sensible heat RMSE is 13.39 $\mathrm{Wm}^{-2}$ with the correlation of 0.82. It should be noted here that sensible heat flux has two different periods during the day, positive when air-water temperature difference is negative, and vice-versa.

More detailed analysis of these curves shows that measurements have minimums of latent heat in the daytime and high peaks

in the evening (20:00 – 21:00). Simulation reproduces these peaks with 1-2 hour delay which are related to the delay on the simulated wind speed. Unfortunately, there is no data about fluxes for 14:00 – 16:00, 22 of July, but according to the model results both fluxes are tend to be around 0 $\mathrm{Wm}^{-2}$.

Wind direction at ALEX stations is represented on the Fig. 8. Measurements show how wind direction changes to the opposite due to the lake breeze effect during the daytime between 21 and 23 of July, not clearly seen on the simulation results

on the grid-points near the ALEX stations. The structure of the simulated breeze will be discussed later. Barbosa station, located on the North-West shore of the lake, indicates the presence of the lake breeze because its direction is the opposite to the dominant wind in this area. However, at Cid Almeida station on the South-East shore breeze is co-directed with the dominant wind in the area, so, its appearance is difficult to track.





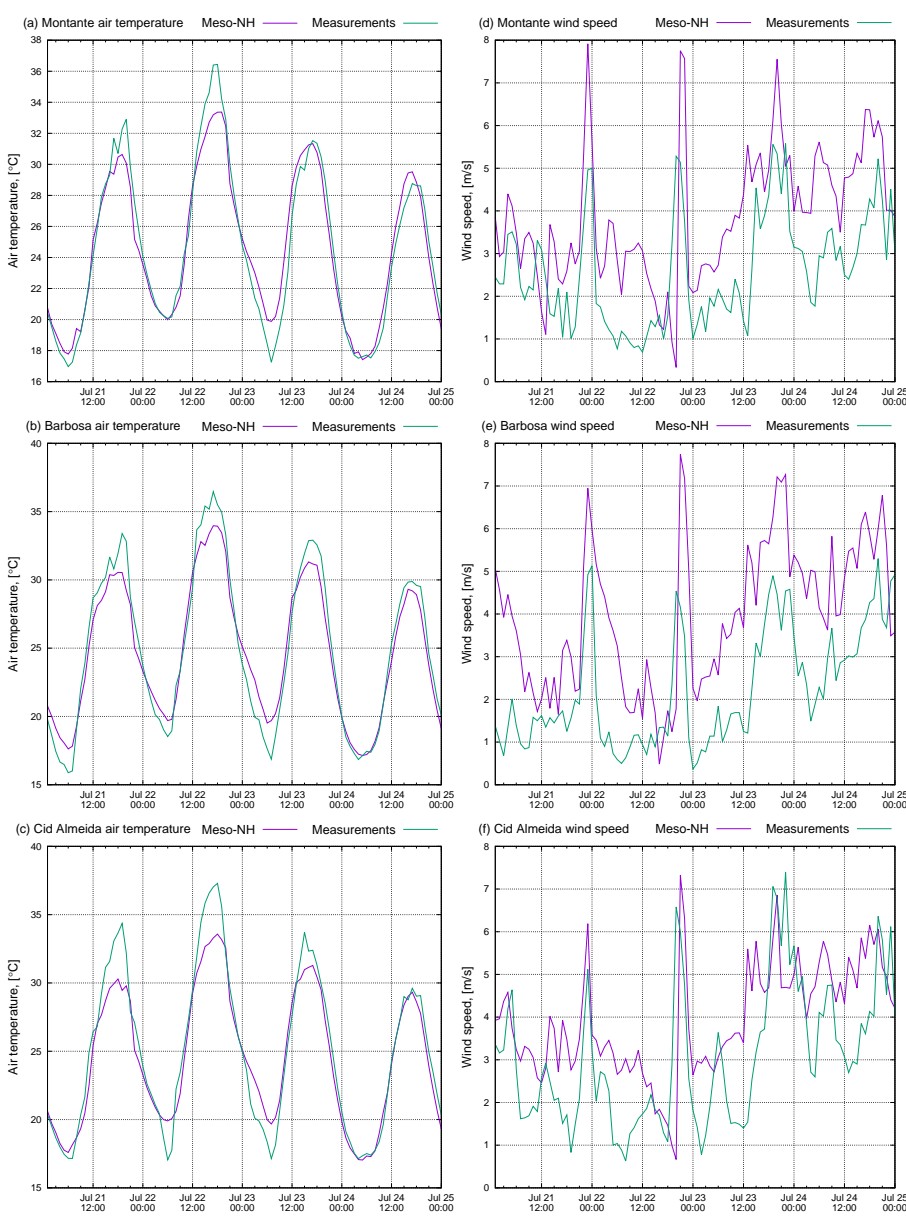

**Figure 6.** Comparison of air temperature and wind speed for ALEX stations: Montante platform (a, d), Barbosa (b, e), and Cid Almeida (c, f) sites.





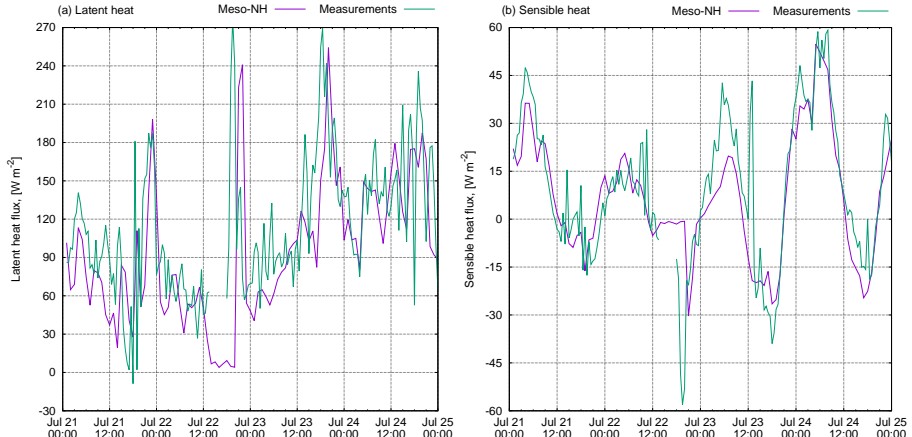

**Figure 7.** Observed and simulated latent (a) and sensible (b) heat fluxes at Montante floating platform.

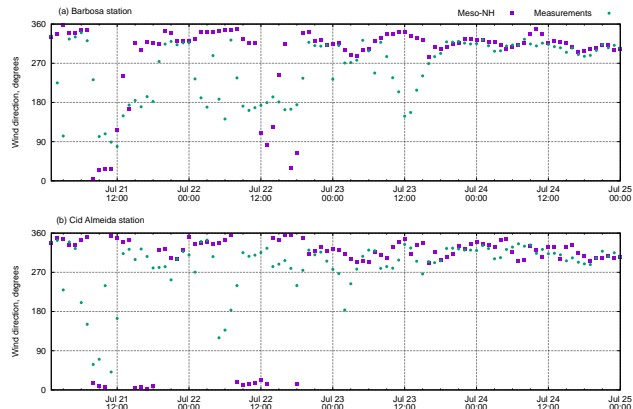

**Figure 8.** Wind direction on Barbosa and Cid Almeida stations.


## 6   Lake effects

To study how the Alqueva lake affects the local area the following atmospheric parameters were used in this work: air temperature and potential temperature, relative humidity and water mixing ratio, and vertical and horizontal wind speed. Overall, simulation result is a 3 sets of 96 output files (for each horizontal resolution) of 1-hour timestep consisting of required atmospheric parameters. Only 1-km and 250-m resolution datasets were used in this section.

During daytime water temperature is lower than air temperature, which is associated to a very weak air circulation over the water surface which leads to very low evaporation from the lake (refer to low latent heat flux values on Fig. 7 (a)). At this time period evaporation over the land is even higher that over the water. By late afternoon when the dominant sea breeze system reaches the region, smooth water surface significantly enhances the North-West wind. As the result, evaporation from the lake becomes very intensive.

Figure 9 shows an example of wind regime on the 1 km domain in the morning (07:00 UTC). Dark blue color represents the lake area over the orography. At this time of the day before the establishment of the lake breeze, North-West wind prevails in the area with the magnitude of 3-5 m/s.

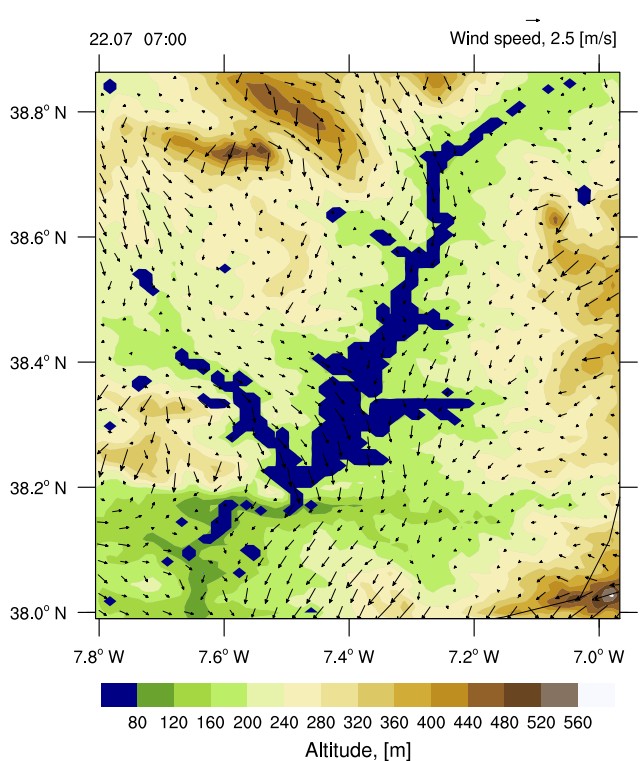

**Figure 9.** Simulated 10 m wind (vectors) over orography (color scale) at 07:00 UT 22 July 2014; results from the 1 km resolution domain.



The first level of air above the lake surface is the most affected by its impact. Fig. 10 illustrates the air temperature difference caused by the presence of the reservoir at 2 meters height during 22 July 2014. We focus on this day because air temperature was the highest and the lake breeze expected to be strong and well distinguishable. Positive anomaly (up to 3-4 K) can be traced during the period from 1 hour after the sunset (21:00 UTC) until 1 hour after the sunrise (07:00 UTC). Examples of this positive

night anomalies are illustrated in Fig. 10 (a,b). Night North-West wind transports warm air from the lake to the South-East part of the reservoir for up to 2 km away from the shore. Daytime period is characterized by the negative temperature anomaly up to 7 K (Fig. 10 (c-f)). This effect is essentially limited by the lake borders. When the large-scale sea breeze system arrives to the Alqueva area, temperature trace of the lake impact are followed by the wind and can be found 10-12 km away from the South-East part of the reservoir (Fig. 10 (f)).

Vertical cross-sections can illustrate the processes in the atmosphere on different altitudes. Such East-West cross-sections along 38.215 °N (Fig. 11, position of this cross-section is indicated by a lower horizontal line on Fig. 10, (a)) show the evolution of wind and the potential temperature during the 22 July in the experiment with the reservoir (simulation LAKE1). Maximum of the temperature impact of the lake can be found in the early afternoon, at 12:00 – 14:00 on the altitudes up to 1-1.2 km, cooling all the boundary layer, which depth decreases (very clear seen on Fig. 11 (a)) from more than 2 km above the surface

outside the zone of influence of Alqueva, to values close to 1 km over the water reservoir. The thermal anomaly induced by the presence of the reservoir seems to affect an area greater than that identified at the surface, especially in the middle of the boundary layer. Later on, at 19:00 – 20:00 the powerful ocean breeze system reaches the area and cools the lower (1 km) layer of air by 6-7 K. The progression of the sea breeze front is impressively well shown on Fig. 11 (d) (20:00 UT), when it reaches the border of the reservoir, and on Fig. 11 (e) (21:00 UT), when it is already beyond the east bank of the Alqueva lake.

Differences in near surface sensible heat fluxes and consequently in air temperature during the daytime induces the formation of the breeze system. The development of the lake breeze is illustrated on Fig. 10. Wind arrows corresponding to speed lesser than 0.5 m/s are not ploted. During nighttime (Fig. 10 (a), (b)) large-scale circulation, driven by the peninsular scale sea breeze system is dominant in the area but after the sunrise when the temperature anomalies near surface changes to negative, winds blowing out of the the lake shores can be observed. Daytime cross-sections on Fig. 11 (a, b, c) indicates that the direct

lake breeze can be found on the altitudes up to 300 meters above lake surface. Breeze wind speed in that case can reach 5-7 m/s. Spreading of the lake breeze in horizontal plane depends on the local orography, but usually the traces of it can be found in 4-6 km away from the lake shores (Fig. 11 (c)). An upper-level convergent return circulation can be noticed (Fig. 11 (a, b, c)) by an increase of eastward component over the west shore, and an westward motion to East of the reservoir. We will return to this features later, during the discussion of the impacts of the teservoir on the moisture field, showing another cross-section,

in which the structure of the lake breeze system is more visible.

This breeze wind intensifes until late afternoon (Fig. 10 (e)) when the wind speed can reach 7 m/s. When the negative temperature anomaly due to the presence of the mass body is getting weaker, breeze system starts to dissipate. At 19:00 – 20:00 when the ocean breeze arrives to the area, lake breeze already can not be traced (Fig. 11 (d, e, f)).

Alqueva impact on the relative humidity at 2 m height is shown on Fig. 12. The lake increases relative humidity up to 50%

during the daytime while at night its influence is insignificant. The positive anomaly is limited by the area of the lake and does



**Figure 10.** 2 m temperature anomaly (difference between LAKE1 and LAKE0 experiments) in filled contours and horizontal wind in LAKE1 experiment (arrays, the scale is indicated in the upper rigth corner of each figure) of the reservoir on 22 of July 2014 on the 250 m resolution domain.





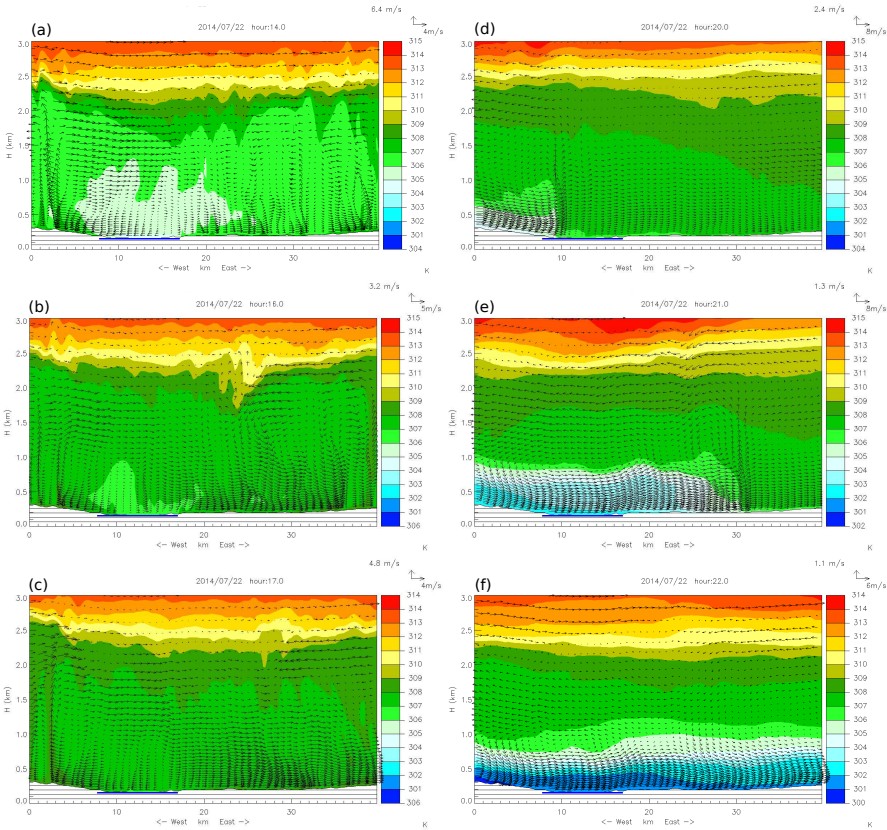

**Figure 11.** East-West direction cross-sectons along 38.215 °N (crosses the lake near Montante platform, southernmost straight line in Fig. 10 (a)) of potential temperature (filled contours) and projection of wind (arrays), at different times (indicated in the top of each figure) in LAKE1 experiment at 250 m horizontal resolution. The wind vertical and horizontal scales are indicated in the upper right corner of each figure. Blue line on the surface level indicates the location of the reservoir.

not spread over the surrounding land. It should be noted that the increase in the relative humidity is mainly due to the decrease in air temperature.

In fact, cross-sections presented on Fig. 13 (the position of the cross-sections are indicated with horizontal lines on Fig. 10 (a)) show that the lake breeze system includes a descending branch over the lake area that carries dry air from a height of about 2-2.5 km and redistribute it over the lake surface. Two different locations of cross-sections (the first one is near Montante platform and the second one is in the middle of the lake) are shown to provide a better vizualization of the three-dimensional structure of air circulation above the lake. This dry downstream effect is confirmed by the results of measurement of water vapor mixing ratio at Montante platform as seen in Fig. 14, in which simulation results are compared with observations: it decreases to a minimum of about 7-8 g Kg$^{-1}$ every day after noon around 14:00-15:00 and reaches a minimum value lower than 6 g Kg$^{-1}$ during the afternoon of the day of stonger lake breeze (July 22). Out of the period in which the air subside over





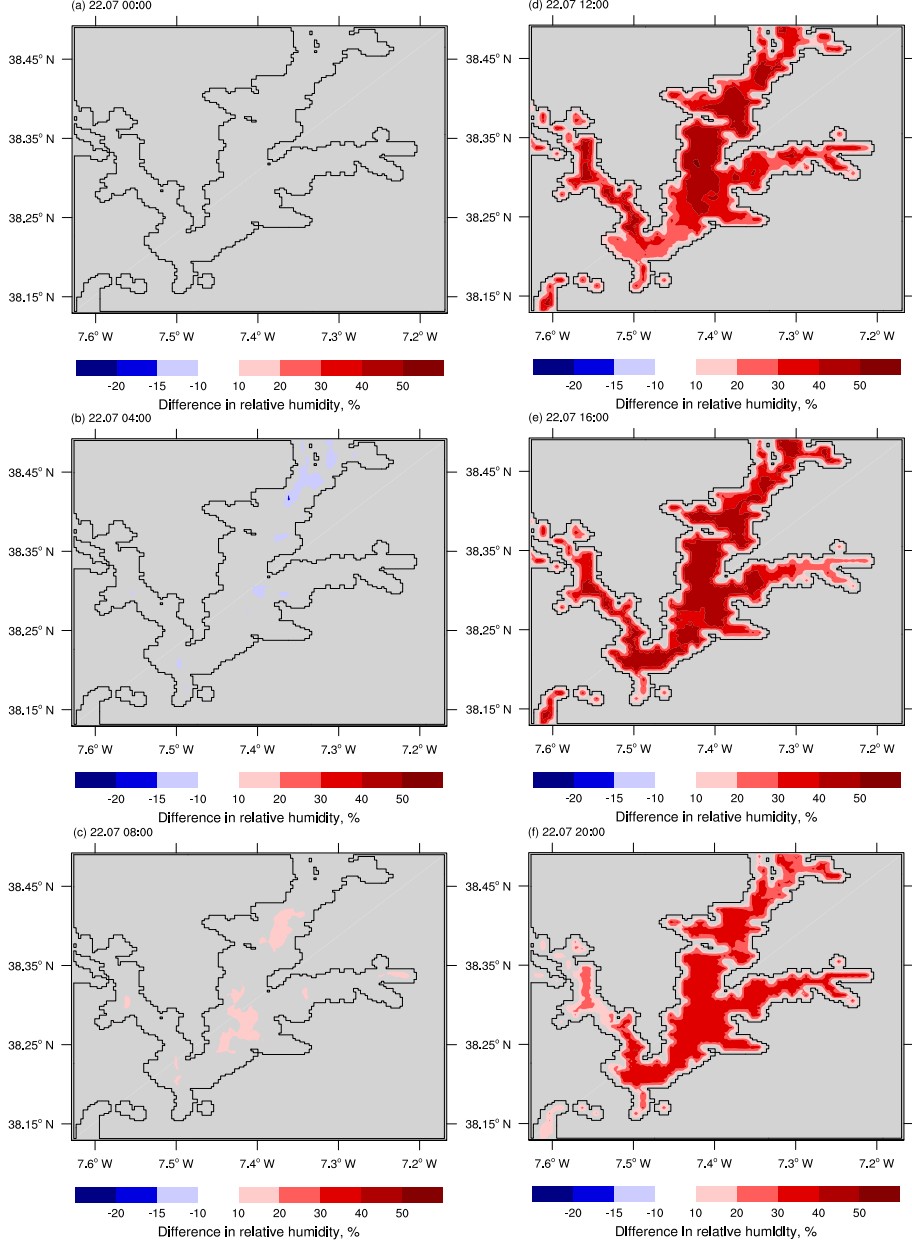

**Figure 12.** 2 m relative humidity anomaly (difference between LAKE1 and LAKE0 experiments) in filled contours on 22 of July 2014 on 250 m resolution domain.

the lake, the mixing ratio returns to previous values of 9-10 g Kg$^{-1}$. The presence of this dry downstream was proposed as a hypothesis by Potes et al. (2017) and is proved through the simulations done in this study. On the other hand, Fig. 13 also shows that outside the reservoir there exist zones of low-level convergence and upward motion which increase the moisture of




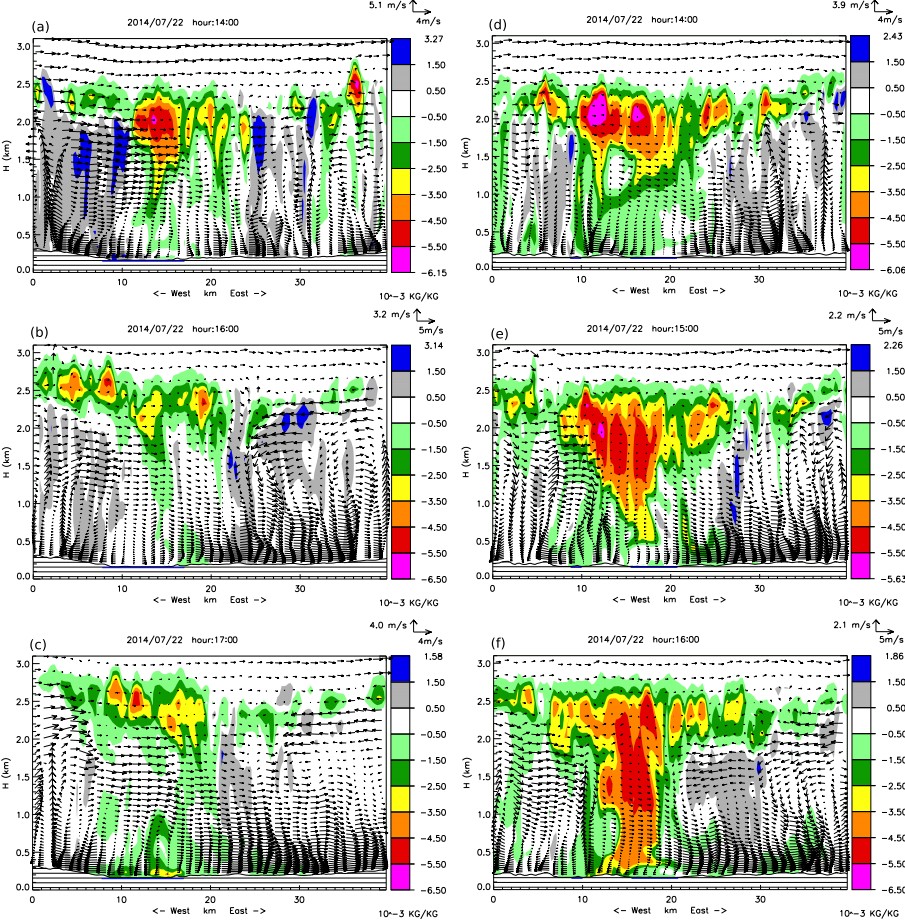

**Figure 13.** East-West direction cross-sections along 38.215 °N (a,b,c) and 38.274 °N (d, e, f) (horizontal lines on Fig. 10 (a)) with the difference (LAKE1 and LAKE0 simulations) of water mixing ratio (filled contours), and projection of wind (arrows) in LAKE1 experiment at 250 m horizontal resolution at different times (indicated in the top of each figure). Blue line on the surface level indicates the location of the reservoir.

the boundary layer. this zones correspond to some kind of lake breeze fronts. The complex shape of the reservoir implies an also complex 3D structure of the breeze system. Towards the Southernmost part, near the dam, the low level divergent breeze circulation is very clear, but the convergence upper-level return current is weaker (Fig. 13 (a, b, and c)). In contrary, near the midle of the reservoir (Fig. 13 (d, e, and f)) where two water branches exist, the circulation near the surface is more complex
5   due to the presence of a land band inbetween, but the subsidence motion is more proeminent, inducing a decrease in mixing ratio through the boundary layer, which reaches a magnitude of about 4 g Kg$^{-1}$ at 16:00 (Fig. 13 (f)).





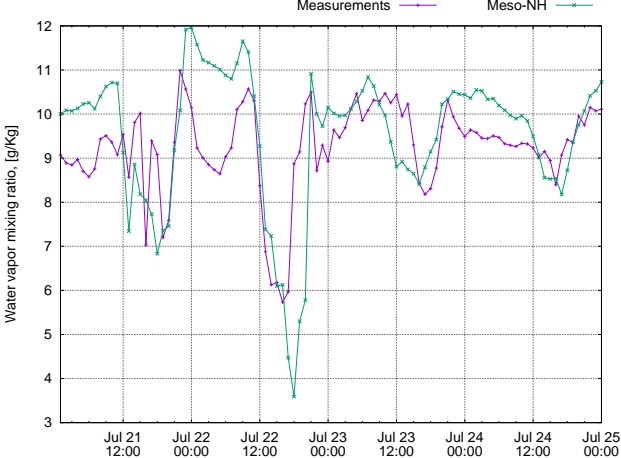

**Figure 14.** Water vapor mixing ratio over Montante platform.

## 7 Conclusions

In this work we studied the formation and magnitude of the summer lake breeze at the Alqueva reservoir, South Portugal, and the impact of the artificial lake on the local weather. The study was based on Meso-NH simulations of an well documented case study of 22-24 July 2014. The model allowed to conduct the simulation with horizontal resolution of 250 meters which is fine enough to figure out such relatively small scale lake breeze and to spot the impact of the reservoir on the local boundary layer structure.

Due to the "youth" of the Alqueva reservoir it is possible to run atmospheric model with the surface conditions prevailing before the filling of the reservoir. Two simulations, one with the Alqueva and another one without it, allow to evaluate the impact.

We described the formation of the lake breeze system during the daytime and its dissipation in late afternoon anticipated by the arrival of the larger scale sea breeze generated at the Portuguese west Atlantic coast. The magnitude of the lake breeze can reach 6 m/s. It can be traced at about 6 km away from the lake shore and on altitudes up to 300 m above the lake surface. Daytime lake regime can be characterized by very low evaporation rate from water surface, while at nighttime major sea breeze induces very high evaporation rate.

Cooling effect of the lake expressed in lower air temperatures (up to 7 K) but limited by the lake borders and normally not seen farther than few kilometers away from the shore mostly in South-East direction. Altitude effect of the cooling can be found at the 1200 m above the lake surface.

The lower layer of the air over the lake usually are more wet during the daytime, but the presence of the lake makes a negative impact on the humidity at higher altitudes. Downward circulation induced by the lake breeze brings dry air from the upper atmospheric layer (2-2.5 km) to near surface levels.





*Competing interests.* The authors declare that they have no conflict of interest.

*Acknowledgements.* The work is co-funded by the European Union through the European Regional Development Fund, included in the COMPETE 2020 (Operational Program Competitiveness and Internationalization) through the ICT project (UID / GEO / 04683/2013) with the reference POCI-01-0145-FEDER-007690 and also through the ALOP project (ALT20-03-0145-FEDER-000004). Experiments were

5   accomplished during the field campaign funded by FCT and FEDER-COMPETE: ALEX 2014 (EXPL/GEO-MET/1422/2013) FCOMP-01-0124-FEDER-041840.





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
