# Peer review of "Breeze effects at a large artificial lake: summer case study"

_Hydrology and Earth System Sciences, 2018_

## Referee Comment (RC1) · Anonymous Referee #1 · 21 May 2018

General comments:

This paper studies the lake breeze effects caused by the Alqueva reservoir (Portugal), which is the largest artificial lake in Western Europe. The paper concentrates to a 3 days long modeling case study done with the Meso-NH model. Simulations are done with and without the reservoir and different kind of measurements are used to evaluate the skill of the model. The results show the existence of a lake breeze and how it influences the local areas. The paper links nicely to previous studies and support their analysis of the breeze effects.

I think the paper fits in the scope of HESS and should be published after some modifications. There are some specific areas that need more analysis and modifications. The language of the manuscript should be improved as there are too many sections

when the text is rather difficult to follow. I have marked some points to the "Technical corrections" section, but the list is not comprehensive. I suggest that the authors get editing help from someone with full professional proficiency in English.

Specific comments:

P1, line 11: Does "It" at the end of the sentence link to the lake breeze or to the Atlantic breeze system? This part is unclear without reading the text.

Figure 1: The text in the a) part is too small. Please consider saying grid boxes instead of pixels. Also, would be more informative if the pictures would actually show the grid boxes, i.e. the resolution would be more visible. The underlying map could be surface orography, like in Figure 9.

The units used in this manuscript seem to have slightly different font than the main text. Is there a reason for this?

P4, lines 14-15: You discuss here about the dataset (the main dataset for this work). It would be clearer to talk about "the measurement data". The modeling data is also a dataset.

P6, lines 15-16: This small chapter could merged with the first chapter of section 4.1. Also, you mention that ECMWF data is used at the lateral boundaries with an update frequency of 6 hours and in chapter 4.1 that the model is capable of doing multi-scale grid nesting techniques. It would be nice to know more how the simulations were really done. I would assume that ECMWF data was used only for the 4-km resolution (even then 6-hourly boundary forcing seems to be a bit coarse) and the higher resolution used some kind of nesting to this (e.g. 1-km was nested to 4-km and 250-m was nested to 1-km). If this is the case, what was the later boundary update frequency of the nests? Overall, more details about the modeling structure are needed.

P7, line 14: Do you have a citation for value used for attenuation coefficient?

P9, lines 5-6: Did you try to include the radiosonde accuracy limits in Figure 4? This

could improve the plots.

P9: You mention the supplementary material, but not the numbers of the figures you are referring to. Please add these to the text.

P10, lines 10-11 and P12, lines 10-11: The comparison of wind speed is interesting in Fig. 4, but what information does it bring to compare the 10-meter wind speed from the model against the measured 2-m wind speed (Figures 5 and 6)? Did you try to convert the 2-m wind speed to 10 m height (or vice versa)? You mention this possibility, but why was it not done? Comparing the same variable on different heights requires more explanation in the text.

P11, Fig. 5. The text font is quite small. Please increase it.

P12, lines 14-15: Are the simulated results more smooth due to difference in plotting frequency (modeled output 1-hourly, what about the measurements? I could not find the information from page 5 for latent and sensible heat fluxes; I assume it is the same as for the variables listed in P5L1). Are the model outputs accumulated over the output frequency? What about measurements? The peak difference seems to be quite large, especially on July 22nd and there should be more discussion about this.

P12, line 14: What are you trying to say with "Dynamic"?

P16, lines 1-9 and Figure 10: Please change the colorbar scale as currently it is too coarse. Perhaps you could try using the limits -5 to 5 degrees with 0.25 degree

P16, line 32: You can use the word "lake" instead of "mass body"

P17, line 34 – P18, line 2 and Fig 12: Like with Fig 10, I think you are using too coarse colorbar in your plots (-10 to 10 % difference are not shown) to see the effect of transport (and night-time differences). Please try to improve the figure in this respect and update the text accordingly.

P17, Fig 12: Could you please name the cross-sections (e.g. I and II) and inform about

this in the caption. Also, please refer to this naming in the text when discussing about the cross-section results.

P18, Figure 11: Could you add to the plot the BL height as seen by the model? Please also increase the font size.

P18, line 9: The water vapor mixing ratio indeed has a minimum around 14:00-15:00 o'clock, but the values are higher than 7-8 g/km on July 23rd and 24th (8-9 g/kg). So the minimum values are not between 7-8 g/kg every day.

P20, line 2: Where is the dam exactly? Please mark it to the maps.

P20, Fig 13: Please increase the font size.

Could you have done any lake water temperature (surface) comparison between the measurements and the model? Although the simulation period is short, the comparison would give some information how good your initial conditions were and how well you model the lake dynamics and the atmosphere-lake interactions.

Conclusions: You list the main results of your work (basically the lake breeze effects), but I would like to see a bit more discussion about their implications.

Technical corrections:

P1, line 1: "could" to "can"

P1, lines 1-2: rewrite the end of the sentence starting from "but usually"

P1, line 2: "lakes" to "lake"

P1, line 5: comma after "reservoir"

P1, line 6: here FLake scheme is used, later FLake model (e.g. P2, lines 26-27). Please be consistent with the description (model is widely used)

P1, line 7: "this" to "these"

P1, line 8: the reservoir

P1, line 18: "0.35 %" to "0.35%"

P1, line 26: "the warm summer period" to "warm summer periods"

P1, line 26: ", forcing" to "leading to"

P2, lines 3-4: Consider changing to "These regional lake effects have been seen in previous studies"

P2, lines 10-11: remove ", and others" and use "and" before "terrain types..."

P2, line 12: "In this work,"

P2, line 14: "A first report" to "The first report"

P2, line 15: "as part of" to "as a part of"

P2, lines 16-19: The sentence starting "They concluded, " is very hard to follow. Please rewrite and make it clearer.

P2, line 20: "were done" to "was done"

P2, lines 22-24: Sentence starting "Later," should be improved

P2, lines 25-28: this chapter needs to be rewritten

P2, lines 29-33: This chapter needs also to be improved. Especially the first sentence and the end of the chapter requires some attention.

P2, line 34: Is "the object of current study" really needed?

P3, line 1: move "used in this work" after "numerical models"

P3, line 7: "if" to "of"

P4, line 5: Consider starting a new sentence with "An average annual..."

P4, line 8: remove "inside it"

P4, line 16: "has last" to "lasted"

P4, line 16: remove "included" and replace it with something like "was to utilize"

P4, line 27: Add comma after "also"

P4, lines 30-31: Consider changing the end to "while the floating platform Montante situated in the middle"

P5, line 9: "has last" to "lasted"

P5, line 10: "have been" to "were"

P5, line 13: consider writing "because the atmosphere was mostly stable and anticyclonic conditions were present"

P5, lines 20-21: Add the first sentence to the first chapter, i.e. remove the gap (line break).

P5, line 22: please add ", and" after "precipitation"

P5, line 24: "platforms" to "platform"

P6, Table 1: "Deep convction" to "Deep convection"

P6, lines 4-6: The sentence starting with "Mixed microphysical..." should be rewritten. A suggestion: "A mixed-phase microphysical sheme...", leave out "and explicit precipitation" and add to the end "was used". The next sentence could start with "The model solves longwave and..."

P6, line 8: "exchange is controlled"?

P6, lines 11-14: The end of the chapter should be improved. For example, for the 1-km and 250m domains it is better to say something like "the resolution is high enough for the deep/shallow convection to be solved explicitly" Also, the reference to Table 1 is

missing the number and the brackets are left open. The list is missing "and" from the end.

P6, lines 15-16: remove "files" and ending should be improved (e.g. "for lateral boundary forcing with an update frequency of 6-hours")

P6, line 17: "the one" to "one"

P6, line 19: remove "-"

P6, line 20: "have covered" to "covered"

P7, line 5: "the freshwater lake..."

P7, line 6: "were" to "was"

P7, line 15: A new chapter starts so better to say "The initial parameters used in FLake are"

P7, line 29: rewrite, a suggestion "the depth of the artificial lakes varies spatially, because"

P8, lines 8: "to compare" to "analyzed"

P8, line 10: "of" to "took place between" (and remove second "took place")

P8, line 13: merge this chapter with the first one in section 5.1

P8, line 14: comma after "point"

P8, line 17: rewrite the end of the sentence, e.g. "22 km height; thus, to build a corresponding profile, three..."

P9, line 6: "95.5 %" to "95.5%"

P9, line 17: "11.26 %" to "11.26%"

P9, line 14: "magnitude are" to "magnitude is"

P9, lines 11 and 15: "suplementary" to "supplementary"

P9, lines 15 onwards: make a real list of the statistical values ("following: temperature average bias. . ., humidity average. . ., and for the wind speeds...")

P10, line 1: "accordance" to "accord"

P10, line 6: "are" is missing. Also, you could add "It should be mentioned that not all..."

P10, line 8: "visible in" should be change to e.g. "which can be seen from"

P10, lines 10-11: this small chapter can be merged with the previous one

P10, line 10: "100 %" to "100%"

P12, line 1: This seems to be a new chapter and yet you referrer to "these stations", please correct.

P12, lines 23-24: Please rephrase this sentence (starting "Measurements").

P15, line 1: Rephrase, e.g. "To study. . . affects the surrounding area, the following. . . were analyzed in this work:"

P15, lines 2-5: Is the sentence starting with "Overall, simulation result..." necessary?

P15, line 6: A comma after "During daytime" and add "the" before water temperature and air temperature.

P16, line 1: A new chapter and you refer with "its" to? Lake breeze should be mentioned here.

P16, lines 12-15: Please rephrase the sentence starting with "Maximum of the temperature" It is too long and complicated.

P16, line 22: "nighttime" to "night-time" and you could move the part in brackets to be before the comma.

P16, line 31: A new chapter starts, where does "this" referrer to?

P18, line 3: use "the cross-sections"

P18, line 9 and Fig. 14: "g/Kg" to "g/kg"

P19, line 3: "increase" to "increases"

P20, line 1: "this zones" to "These zones"

P20, line 5: "proeminent" to "prominent"

P20, Figure 13: "KG/KG" to "kg/kg"

P21, line 5: "figure out" to "resolve"

---

## Referee Comment (RC2) · Anonymous Referee #2 · 23 May 2018

This paper investigates changes in atmospheric variables in the area of Lake Alqueva, induced by the filling of this artificial lake in 2004. To identify the changes, two simulations were performed using a mesoscale atmospheric model, the Meso-NH model. In the first experiment, the lake is not present and in the second one, a lake model, Flake, is run in a coupled mode. The authors observed the formation of a lake breeze in the presence of the reservoir and identified impacts on the atmosphere.

This study is interesting as it quantifies the effects of a large lake on the weather of the region. The results are nice and innovative, in particular results presented in Fig. 11 and Fig. 13, but I think the author could go a bit further and relate their findings (in terms of simulations) with changes that have been observed at the weather stations. Did they also notice changes in the observed wind regime between 2010-2018 and

year1990-2000 for instance? Otherwise, the paper looks more like a first draft, which makes the reading quite painful. Some explanations are too vague, some acronyms are not defined, and many sentences are awkward. I highly recommend that an English speaker reads the manuscript before resubmitting.

Specific comments

- The formation of the lake breeze is not clearly explained.

- Some acronyms are given but they first need to be explained. For instance, in the introduction, you mention NH3D. What king of model is it (ex: atmospheric model)? Same, when you mention Meso-NH model, SURFEX, Flake. As well, p.3: what is Csa? (Mediterranean climate) should appear in the text, and Csa should be in brackets (Csa according to the Köppen climate classification). Again, in p.6 ECOCLIMAP and SRTM. You need to clarify.

-P7. L34: You mention Flake results based on 2-4 months simulations. Did you perform these simulations? Which period did you choose to run these simulations? What is the correlation between simulated and observed data? I would like to see how well the model reproduces the surface temperatures. This is very important to assess the intensity of a lake breeze and the accuracy of the results.

-The discussion on the lake effects focuses on the southern part of the lake. Are the conclusions also valid in the Northern part of the lake?

-You mention that changes in relative humidity are mostly related to change in temperature. However, looking at Fig. 10 and Fig. 12, differences do not appear at the same place. It is maybe related to the fact that the hours on each subplot of figures 10 differ from those on figure 12. It would make sense to have something more homogeneous. Also, wouldn't it be worth adding a map, such as Figs. 10 and 12, representing surface specific humidity? Are they several descending branches of dry air over the lake?

-P.12 You indicate the maximum error in terms of temperature. My feeling is that a

bias of 5° is quite a lot and especially when it last for several hours. I would expect a discussion on the impact of this bias on the turbulent fluxes or some hypothesis in order to explain why the fluxes are so well reproduced considering this bias. This could affect modelled lake surface temperature and the intensity of lake breezes.

The lake effect part is very interesting, but it is hard to follow the mechanism you describe. On Figure 11, you should draw circle where you identify "the upper-level convergent return circulation". The figure needs to be bigger.

In the conclusion, I would expect some general comments on your findings. Are the conditions on July 22-24, representative of the conditions that prevail in this area in summer? What kind of experiments should be done in the future or is there anything you would like to investigate further? What are the limits to your conclusion? There are some biases in the atmospheric variables between modelled and observed data. How confident are you in your results?

The units are not systematically the same. Temperatures unit are for instance in °C in Fig. 5 but in K in Table 2.

P5: you mention 3 domains, A, B, and C, why don't you use these terms later in the text? For instance on P6: Domaine B required deep convection... It would make the manuscript easier to read.

Some figures are too small. For instance, Figure 5. Also use the same symbol for corresponding stations on each subplot. Figure 11 needs to be bigger.

Figure 10: you should name the cross section. For instance S1 and S2 and refer to them in the text. That would ease the reading.

Figure 13 and others: it is weird to have different scales for the windspeed. It is then difficult to assess the evolution of the windspeed throughout the day.

P8. You say twice that the domain B is used for validation with radiosondes.

[Figure]

In the dataset section, try to gather the information per station. Also later in the text (p.11), you define the coordinates of the stations.

These sentences are unclear, please clarify:

- P1. Abstract: you say that two simulations have been done with the meso-NH model coupled to Flake. Only one was coupled, no?

- P1: L 25: daily air temperature near the surface is decreased in lake shore areas -> and above the lake?

- P1. L26: lake surface balances the atmosphere above ->clarify

- P2. L3: In autumn and winter it has the opposite effect due to the warmer air above lake surface: increase of evaporation and cloud formation -> not warmer air above the lake in summer?

- P2. L32: Simulation has been done for… -> which simulation are your talking about? A simulation performed within the ALEX2014 experiment?

- P4, L24: 3 stations of Instituto de Ciencias etc… -> what kind of stations? What kind of variables?

- ALEX and ALEX2014, is it the same database?

- P4. The two land stations your refer to, are they the weather stations you mentioned earlier? Gather the information and be consistent. Alqueva Montante and Montante, the same?

- You say that the choice of your study period is based on atmospheric conditions. But you also say that the project lasted for 3 days. Wasn't the choice more based on the availability of data?

- P. 9: the worse values are in the lower lever. What do you mean? Extremely bad?

- p. 9: patterns look similar. Are they similar or do they just look similar?

[Figure]

- P.16: The first level of air above the lake is the most affected by its impact- > impact of what?

- P16: Need to clarify where you mention positive or negative anomalies. Over the lake?, over the land surface?.

- P.18, legend: what do you mean with projection of wind, same for figure 13.

- P.18: the fact RH is decreasing due to change in temperature is an important point. Remove "it should be noted"

Technical corrections

Here is a list of sentences that need to be rephrased :

- P.2 the using of coarse spatial resolution observations data

- On many pages: meteorological variables instead of parameters

- p.2, L20. Remove "in his PhD thesis also in Portuguese", not relevant

- p2, L 27. Surface models Masson et al used among atmospheric models by Meso-NH

- p2. L 29: allows to gain the results

- Many times, you use ":", make a sentence that includes what follows. For instance, you could replace the ":" by "such as "

- P4: L3: "on the surface level? –> at the surface

- P5: L10: For that, ->remove.

- P. 6:Longwave and shortwave radiative transfer equations are solved for independent air columns

- P6: A set of two numerical simulations were performed. . .

- P.9, L15: Temperature average bias is -0.13 K, RMSE 1.49 K, and correlation coeffi-
cient is 0.99. Humidity average bias is 0.59% RMSE of 11.26 % and 0.87 correlation coefficient.

- P10. Scatter plots of air temperature, relative humidity, and wind speed shown on Fig. 5 -> verb missing

- P.10: The worse result are observed in comparison against Portalegre data

- P.11; Legend needs to be clarified. Comparison of modelled air temperature…with

- -p.11. L 7-8, suggestion: In the case the meteorological stations were located in a lake grid cell, the nearest land …

- P12. L2-3: Meso-NH underestimation of air temperature in the afternoon time is opposite of wind speed overestimation at the same period.

- P15, L3: is a 3 sets of -> consist of

- P15, L3: (for each horizontal resolution) > (one set per domain)

- P16, L14: which depth decrease (very clear seen on Fig.11). I don't know what you mean.

Typing errors and other mistakes

- p2. L20: a first attempt were done

- p3. L7: "if" instead of "of"

- p3, L8: 92 m instead of 92.0 m

- p4. L31: locaton

- p6. Table: convction

- p7: Flake were used

- p9, L5: accurace

- p9. L11 et L15: supplementary

- p.9 Statistical results are following

- p. 11: meteostation

- p. 12, L19: minimums

- p.12, L22: are tend to

- p.20, L1: this zone -> This zone

- p.20, L4: midle

- P.10 L14: lesser

- P.10 L13. Verb missing

- P.16, L29: the teservoir

- P.16, L31: intensifes

- P.21: more wet

- P.18: Legend: cross-sectons, at different times-> hours

-

---

## Author Comment (AC1) · 28 Jul 2018

**Breeze effects at a large artificial lake: summer case study**

AUTHORS' RESPONSES TO THE REVIEWER 1 COMMENTS

Maksim Iakunin[1], Rui Salgado[1], Miguel Potes[1]
*miakunin@uevora.pt*

[1]Department of Physics, ICT, Institute of Earth Sciences, University of Évora, 7000 Évora, Portugal

**Contents**

**Introduction. Document structure**

This document contains authors' responses to the comments of the Reviewer. The document structure is the following:

- Reviewer's comments are numbered and given in *italic font*. General, specific, and technical comments come separately.

- Authors' response follows the comment and starts after **"Response:"** with normal font.

- The text from the article itself (if some changes were done, and if it is reasonable to provide it) is typed with `typewriter font` and separated from the response with an extra blank line.

- *Technical comments and mistakes* are not numbered, and authors' response follows immediately.

Reviewed manuscript with all the corrections is given after all responses. It contains the changes and proposals of **two** Reviewers and was prepared using LaTeXdiff package for better understanding of what was added or removed.

**Anonymous Referee 1**

**General comments**

*This paper studies the lake breeze effects caused by the Alqueva reservoir (Portugal), which is the largest artificial lake in Western Europe. The paper concentrates to a 3 days long modeling case study done with the Meso-NH model. Simulations are done with and without the reservoir and different kind of measurements are used to evaluate the skill of the model. The results show the existence of a lake breeze and how it influences the local areas. The paper links nicely to previous studies and support their analysis of the breeze effects. I think the paper fits in the scope of HESS and should be published after some modifications. There are some specific areas that need more analysis and modifications. The language of the manuscript should be improved as there are too many sections when the text is rather difficult to follow. I have marked some points to the "Technical corrections" section, but the list is not comprehensive. I suggest that the authors get editing help from someone with full professional proficiency in English.*

**Response:**   We thank the Reviewer for the positive comments about the text. The paper was edited very carefully and modifications and improvements were made. Below, we address every comment and explain the corresponding changes in the manuscript.

**Specific comments**

**Comment 1**

*P1, line 11: Does "It" at the end of the sentence link to the lake breeze or to the Atlantic breeze system? This part is unclear without reading the text.*

**Response:**   Indeed, after mentioning two different breeze systems in previous sentence, `"it"` here looks confusing. To avoid this the sentence was rewritten:

5    The descending branch of the lake breeze circulation brings dry air from higher atmospheric layers (2-2.5 km) and redistributes it over the lake.

**Comment 2**

*Figure 1: The text in the a) part is too small. Please consider saying grid boxes instead of pixels. Also, would be more informative if the pictures would actually show the grid boxes, i.e. the resolution would be more visible. The underlying map could be surface orography, like in Figure 9.*

10   **Response:**   Font size for towns was increased on Fig. 1 (a) as well as the figure itself was enlarged. We were trying to add some extra layer to provide more information about the domain, but in this case locations of stations become difficult to read, and this does not make figure more useful. Same problem appears if we add grid to the maps: grid points are small and lines appear to be too dense making the figure uninformative. So we decided to keep all three figures (a, b, c) in 15   same style.

[Figure]

Following the suggestions of the reviewer, the caption of the figure was changed to:

Nested domains used in the simulations: (a) Father domain at 4 km horizontal resolution 20   with $100 \times 108$ grid points, with location of the 12 IPMA synoptic stations used for validation, (b) intermediate 1 km horizontal resolution domain, $96 \times 72$ grid points, (c) finer 250 m

resolution domain comprising $160 \times 160$ grid points, together with the location of the ALEX land stations, the Montante floating platform and the dam.

Pixels were replaced by grid points.

**Comment 3**

*The units used in this manuscript seem to have slightly different font than the main text. Is there a reason for this?*

**Response:** Indeed, the font of the units looks slightly different. The reason for this is that we used a HESS LᴬTEX template where indicated:

```
%%% PHYSICAL UNITS
%%% Please use \unit{} and apply the exponential notation
```

This command could make unit fonts look a little bit different.

**Comment 4**

*P4, lines 14-15: You discuss here about the dataset (the main dataset for this work). It would be clearer to talk about "the measurement data". The modeling data is also a dataset.*

**Response:** That is true, and we agree that speaking of measurement data we named them "main dataset" which is not correct in this context. Later we introduce the results of modeling which are part of the dataset as well as measurements. To make it clearer, section 3 was renamed to **Measurement data**, and corresponding corrections in that section was made.

**Comment 5**

*P6, lines 15-16: This small chapter could merged with the first chapter of section 4.1. Also, you mention that ECMWF data is used at the lateral boundaries with an update frequency of 6 hours and in chapter 4.1 that the model is capable of doing multi-scale grid nesting techniques. It would be nice to know more how the simulations were really done. I would assume that ECMWF data was used only for the 4-km resolution (even then 6-hourly boundary forcing seems to be a bit coarse) and the higher resolution used some kind of nesting to this (e.g. 1-km was nested to 4-km and 250-m was nested to 1-km). If this is the case, what was the later boundary update frequency of the nests? Overall, more details about the modeling structure are needed.*

**Response:** Agreed, this small paragraph fits better in the end of the section's **4.1** first paragraph. As for nesting technique. The model is able to conduct simulations on various spatial scales. Since we are interested in atmospheric processes in particular region on the background of some large-scale processes, we can use grid nesting. We set "father" domain to take the larger peninsular scale processes into account and put smaller "son" domain with higher spatial resolution inside. The model runs using a two-way grid nesting technique, in which the results of the simulation in "son" domain are used as a forcing back in the "father's" domain. Each "son" domain may have its own "son" domains, in Meso-NH the "depth" of nested domains is limited to eight.

ECMWF analysis are used to initialise the model and provide up-to-date information about the atmospheric conditions on the boundaries. This is working only for the "father" domain because for all "sons" domains initial and boundary conditions are calculated by the model itself. This information is interpolated from the results of the "father" domain simulation on every timestep. In this work, we use only analysis as boundary conditions and not forecasts, and ECMWF provides analysis only every six hours.

Corresponding additions were made to the paragraph:

In this work, three nesting domains were used: $400 \times 432$ km$^2$ domain with 4 km horizontal resolution to take into account the large scale circulations, namely the influence of the sea breeze (Fig. 1 (a)), an intermediate $96 \times 72$ km$^2$ domain with 1 km horizontal resolution centered at the Alqueva reservoir (Fig. 1 (b)), and a finer $40 \times 40$ km$^2$ domain with 250 m spatial resolution to track the small scale effects of the lake (Fig. 1 (c)). Hereinafter we denote this three domains A, B, and C correspondingly. The two-way nesting technique used in Meso-NH allows to conduct simulations on different horizontal resolutions at the same time. Domain A is a "father" domain for B, which means that simulation results on domain A are interpolated and used as initial and boundary conditions for domain B. Same scheme applies for domains B/C. European Centre for Medium-Range Weather Forecast (ECMWF) operational analyses, updated every six hours, were used for Meso-NH initialization and domain A boundary forcing.

**Comment 6**

*P7, line 14: Do you have a citation for value used for attenuation coefficient?*

**Response:**   Yes, thank you for noticing that. Citation was added (Potes et al., 2017).

**Comment 7**

*P9, lines 5-6: Did you try to include the radiosonde accuracy limits in Figure 4? This could improve the plots.*

**Response:**   Yes, we tried, but it does not improve the plots. Errors in air temperature and wind speed are too small in comparison with the ranges of these variables, so errorbars in this case are not useful. As for relative humidity, it changes very rapidly in lower lower layer of the atmosphere, so errorbars only make plot less informative. You can see that on the example below:

[Figure]

**Comment 8**

*P9: You mention the supplementary material, but not the numbers of the figures you are referring to. Please add these to the text.*

**Response:**   That is true, references to these figures in supplementary materials were missed. Now corresponding references were added.

**Comment 9**

*P10, lines 10-11 and P12, lines 10-11: The comparison of wind speed is interesting in Fig. 4, but what information does it bring to compare the 10-meter wind speed from the model against the measured 2-m wind speed (Figures 5 and 6)? Did you try to convert the 2-m wind speed to 10 m height (or vice versa)? You mention this possibility, but why was it not done? Comparing the same variable on different heights requires more explanation in the text.*

**Response:**    Yes, we did the interpolation of the model output data from 10 meters to 2, which slightly improved the results of the comparison (correlations and biases). Sections **5.2** and **5.3** were revised in accordance with the new results.

**Comment 10**

5      *P11, Fig. 5. The text font is quite small. Please increase it.*

**Response:**    Font size of the legend and axis labels was increased, now the figure should be more readable.

[Figure]

**Comment 11**

10      *P12, lines 14-15: Are the simulated results more smooth due to difference in plotting frequency (modeled output 1-hourly, what about the measurements? I could not find the information from page 5 for latent and sensible heat fluxes; I assume it is the same as for the variables listed in P5L1). Are the model outputs accumulated over the output frequency? What about measurements? The peak difference seems to be quite large, especially on July 22nd and there should be more* 15   *discussion about this.*

**Response:**    Yes, model output data is 1-hourly, but it is not accumulated over this period. Measurements of latent and sensible heat fluxes were done with 30-minutes timestep (information about this was added to page 5, where we speak about Irgason eddy-covariance system). For the validation, measurement data was 1-hourly averaged (this applies to all data, both with 1-minute 20   and 30-minutes timestep), so the timestep was equal. The modelled curve is more smooth for the reason that the model itself is more conservative and usually prevents variables from quick changes.

The paragraph with the discussion of fluxes comparison was expanded:

25      More detailed analysis of Fig 7 (a) shows that the lowest heat flux values usually occur during the afternoon (12:00 − 18:00 UTC), under windless conditions, and high peaks in the early evening (20:00 − 21:00 UTC). The simulation reproduces these peaks with 1-2 hour delay which are related to the delay on the simulated wind speed. The magnitude of the latent heat flux daily maximum (order of 200 − 250 Wm$^{-2}$) is well captured by the model. 30   The delay in the simulation of the peaks reduces the value of the correlation coefficient and is a manifestation of the so-called double-penalty that penalize high-resolution model scores. As seen in the Fig. 7 (b) the simulated latent heat flux is almost zero between 14:00 and 16:00

UTC of July 22. As pointed before, there is a gap in the measurements of the flux during this period, but data from the day before indicates that the results are realistic. This effect of almost zero evaporation from water on a very hot day is contrary to common sense and will be discussed later.

**Comment 12**

*P12, line 14: What are you trying to say with "Dynamic"?*

**Response:** In this sentence the word "Dynamic" should be interpreted as "temporal evolution". The sentence was changed to:

The temporal evolution of simulated and observed latent and sensible heat fluxes. . .

**Comment 13**

*P16, lines 1-9 and Figure 10: Please change the colorbar scale as currently it is too coarse. Perhaps you could try using the limits -5 to 5 degrees with 0.25 degree.*

**Response:** Colorbar stride was reduced to 0.5 and the range changed to $-6.5 \ldots 6.5$. Now the temperature anomalies are more detailed. Reducing the colorbar stride to 0.25 makes the internal structure of the thermal impact too smooth.

[Figure]

**Comment 14**

P16, line 32: You can use the word "lake" instead of "mass body".

**Response:** Yes, "mass body" was replaced by "lake".

**Comment 15**

P17, line 34 – P18, line 2 and Fig 12: Like with Fig 10, I think you are using too coarse colorbar in your plots (-10 to 10 % difference are not shown) to see the effect of transport (and night-time differences). Please try to improve the figure in this respect and update the text accordingly.

**Response:** As well as in the response to *Comment 13*, colorbar was improved. Unfortunately, night-time differences are too weak to trace. Only at midnight (and at 1-2 a.m.) some negative impact can bee seen (Figure below, (a)), and its magnitude is not higher than 20%, while daytime anomalies does not differ noticeably. Corresponding updates to the article were made (section **6 Lake impact**).

[Figure]

**Comment 16**

*P17, Fig 12: Could you please name the cross-sections (e.g. I and II) and inform about this in the caption. Also, please refer to this naming in the text when discussing about the cross-section results.*

**Response:** Yes, that makes sense. Cross-sections are named S1 and S2, and corresponding corrections and references are made in section **6 Lake impact**.

*Comment 17*

*P18, Figure 11: Could you add to the plot the BL height as seen by the model? Please also increase the font size.*

**Response:** In Meso-NH version 5.3.0 (which we used in this work) it is a known issue that boundary layer top is not calculated correctly. We tried to put it to the figures (as you can see on the example below), but as we are not sure of the results, we decided to keep this figure without it.

[Figure]

*Comment 18*

P18, line 9: The water vapor mixing ratio indeed has a minimum around 14:00-15:00 o'clock, but the values are higher than 7-8 g/km on July 23rd and 24th (8-9 g/kg). So the minimum values are not between 7-8 g/kg every day.

**Response:** True, these values were revised, and then the part of paragraph was rewritten:

This dry downstream is confirmed by the measurements of water vapor mixing ratio at the Montante platform. As can be seen in Fig. 13 the observed and the simulated mixing ratio of water vapor have a daily minimum with average values of about 8-8.5 g kg$^{-1}$ around 14:00-16:00. During the afternoon of July 22, the day with a strong lake breeze, the minimum reached a value lower than 6 g kg$^{-1}$. Out of the period in which the air over the lake subsides, the water vapor mixing ratio returns beck to 9-10.5 g kg$^{-1}$. The presence of this dry downstream was proposed as a hypothesis by Potes et al. (2017) and is proved through the performed simulations. In the same Fig. 13 it is clearly seen that the model tends to overestimating the mixing ratio, except in the afternoon of July 22.

*Comment 19*

P20, line 2: Where is the dam exactly? Please mark it to the maps.

**Response:** The location of the dam now is mentioned in the end of the first paragraph of section **2 Object of study**:

The dam is located in the southern part of the reservoir (Fig 1 (c)).

Corresponding adjustments are made in Fig. 1 (see *Comment 2*).

*Comment 20*

P20, Fig 13: Please increase the font size.

**Response:** Figure was enlarged:

[Figure]

**Comment 21**

*Could you have done any lake water temperature (surface) comparison between the measurements and the model? Although the simulation period is short, the comparison would give some information how good your initial conditions were and how well you model the lake dynamics and the atmosphere-lake interactions.*

**Response:** Yes, we performed this comparison. The results added to Fig. 2. Also the corresponding text added to the end of **4.2** section:

[Figure]

The comparison between measurements of water temperature near surface (at 1 meter depth) and FLake simulated values of mixed layer temperature are shown in Fig. 2 (b). Sensor at 1 meter depth was chosen because it always stays in mixed layer and is not affected by surface "skin" effects. Modelled values are close to measurements which indicates that the initial conditions were realistically imposed.

**Comment 22**

*Conclusions: You list the main results of your work (basically the lake breeze effects), but I would like to see a bit more discussion about their implications.*

**Response:**    Section **Conclusions** was expanded and more comments were added:

This work is dedicated to the studies of the formation and magnitude of the summer lake breeze at the Alqueva reservoir, South Portugal, one of the impacts of the artificial lake on the local weather. The study was based on Meso-NH simulations of a well documented case study of 22-24 July 2014. This period was taken for several reasons. First, a large volume of meteorological data was collected during these days, which allowed for a validation of the simulation results. Secondly, this period was hot and dry, which is typical for most summer days in this region.

The model allowed to conduct the simulation with horizontal resolution of 250 meters which is fine enough to resolve such relatively small scale lake breeze and to spot the impact of the reservoir on the detailed local boundary layer structure. Due to the "youth" of the Alqueva reservoir it is possible to run atmospheric model with the surface conditions prevailing before the filling of the reservoir. Two simulations, one with Alqueva and another one without it, allow to evaluate the raw impact of the lake on the local weather regime.

Formation and dissipation of the daytime breeze system induced by the reservoir are described in the work. On hot summer mornings the difference between air temperatures above water and neighbouring land surfaces induces the radial movement of air from the lake. The breeze system starts to form in the morning and the peak of the wind speed reaches 6 m/s in the late afternoon. Simulation results show that the lake breeze could be detected at a distance of more than 6 km away from the the shores and on altitudes up to 300 m above water surface. In late afternoon the dissipation stage of the lake breeze system anticipated with the arrival of the larger scale sea breeze from the Portuguese west Atlantic coast. In early evening (19:00 − 20:00 UTC) the local lake breeze system can not be detected anymore. No reverse land breeze is detected during the night.

During daytime, the simulation testify the observed very low evaporation from water surface ($0 - 120$ Wm$^{-2}$ in terms of sensible heat flux), due to weak winds and the stable stratification of the internal atmospheric surface layer. A night-time, the strong winds associated with the Peninsular larger-scale circulation induced by the sea-land contrasts, induce a very high evaporation rate ($200 - 250$ Wm$^{-2}$).

The cooling effect of the reservoir can decrease the air temperature up to 7 °C, nevertheless is limited by the lake borders and normally can not be seen farther than few kilometers away from the shore mostly in southeast direction. The cooling can be found up to 1200 m above the lake surface.

Lake breeze system brings dry air from upper atmospheric layers (2-2.5 km) to near surface levels above the reservoir. This effect leads to the fact that the air above the surface of the lake becomes more dry in terms of water vapor mixing ratio, in spite of its relative humidity can increase up to 50% due to the decrease in air temperature.

Further work implies two directions. The first is tuning the lake model and its initialization in order to obtain more accurate results and reduce validation biases. The second is to carry out a longer experiment, which would cover a 12-month period. Such simulation could reveal seasonal aspects of the impact of Alqueva on local weather.

**Technical comments**

*P1, line 1: "could" to "can"*
Corrected.

*P1, lines 1-2: rewrite the end of the sentence starting from "but usually".*
The end of the sentence was rewritten:

Natural lakes and big artificial reservoirs can affect the weather regime of surrounding areas but, usually, consideration of all aspects of this impact and their quantification is a difficult task.

*P1, line 2: "lakes" to "lake"*
Corrected.

*P1, line 5: comma after "reservoir"*
Added.

*P1, line 6: here FLake scheme is used, later FLake model (e.g. P2, lines 26-27). Please be consistent with the description (model is widely used).*
Corrected, now "model" is used throughout the text.

*P1, line 7: "this" to "these".*
Corrected.

*P1, line 8: the reservoir.*
Corrected.

*P1, line 18: "0.35 %" to "0.35%".*
Corrected.

*P1, line 26: "the warm summer period" to "warm summer periods".*
Corrected.

*P1, line 26: ", forcing" to "leading to".*
The sentence was rewritten:

During the warm summer periods relatively colder lake surface interacts with the atmosphere above, which leads to a reduction of clouds and precipitation.

*P2, lines 3-4: Consider changing to "These regional lake effects have been seen in previous studies".*
Replaced.

*P2, lines 10-11: remove ", and others" and use "and" before "terrain types...".*
Removed and replaced.

*P2, line 12: "In this work,"*
Corrected.

*P2, line 14: "A first report" to "The first report".*
Corrected.

*P2, line 15: "as part of" to "as a part of".*
Corrected.

*P2, lines 16-19: The sentence starting "They concluded, " is very hard to follow. Please rewrite and make it clearer.*
The sentence was split into two and rewritten:

It was concluded that the climate impact of the multi-purpose Alqueva project should be merely due to the irrigation of surrounding area. The influence of the reservoir itself was unclear as at that time it was not possible to perform high resolution simulations.

*P2, line 20: "were done" to "was done".*
Corrected.

*P2, lines 22-24: Sentence starting "Later," should be improved.*
The sentence was rewritten:

Later on, Policarpo et al. (2017), used observations data from two periods of ten years (before and after Alqueva reservoir) combined with Meso-NH simulations, and showed a slight increase in the average number of days with fog during the winter (about 4 days per winter after 2003 in a downwind site)

*P2, lines 25-28: this chapter needs to be rewritten.*
This paragraph was removed from the article.

*P2, lines 29-33: This chapter needs also to be improved. Especially the first sentence and the end of the chapter requires some attention.*
The paragraph was rewritten:

Mesoscale atmospheric models, such as Meso-NH, allow to obtain results with sufficient horizontal resolution (250 m in present study) for studying the local effects of air temperature changes and the generation of small-scale circulations under different large-scale atmospheric situations. In this work simulations have been done for the Intensive Observation Period (IOP) of ALEX project (ALqueva hydro-meteorological EXperiment, http://www.alex2014.cge.uevora.pt/). Data collected during this experiment were used to validate the numerical simulations.

*P2, line 34: Is "the object of current study" really needed?*
Not really. Removed.

*P3, line 1: move "used in this work" after "numerical models".*
Corrected.

*P3, line 7: "if" to "of".*
Corrected.

*P4, line 5: Consider starting a new sentence with "An average annual...".*
Rewritten:

The normal (1981-2010) average annual precipitation in the city of Beja (40 km from Alqueva reservoir) is 558 mm (www.ipma.pt).

*P4, line 8: remove "inside it".*
Removed.

*P4, line 16: "has last" to "lasted".*
Corrected.

*P4, line 16: remove "included" and replace it with something like "was to utilize".*
The sentence was rewritten:

One of the aims of this project was to perform a wide set of measurements of chemical, physical, and biological parameters in the water, air columns, and over the water-atmosphere interface.

*P4, line 27: Add comma after "also".*
Added.

*P4, lines 30-31: Consider changing the end to "while the floating platform Montante situated*

*in the middle".*
Changed.

5      *P5, line 9: "has last" to "lasted".*
Corrected.

*P5, line 10: "have been" to "were".*
Corrected.

10      *P5, line 13: consider writing "because the atmosphere was mostly stable and anticyclonic conditions were present".*
Rewritten:

This period, 22-24 July, was chosen for a case study in the this work,as it is an well docu-
15 mented period with typical anticyclonic conditions, hot, dry and low near surface wind speed.

*P5, lines 20-21: Add the first sentence to the first chapter, i.e. remove the gap (line break).*
Removed. Now it is one paragraph.

20      *P5, line 22: please add ", and" after "precipitation".*
Added.

*P5, line 24: "platforms" to "platform".*
Corrected.

*P6, Table 1: "Deep convction" to "Deep convection".*
Corrected.

*P6, lines 4-6: The sentence starting with "Mixed microphysical..." should be rewritten. A*
30 *suggestion: "A mixed-phase microphysical sheme...", leave out "and explicit precipitation" and add*
*to the end "was used". The next sentence could start with "The model solves longwave and...".*
Sentences were rewritten:

A mixed-phase microphysical scheme for stratiform clouds and explicit precipitation (Co-
35 hard and Pinty, 2000; Cuxart et al. 2000)) which distinguishes six classes of hydrometeors
(water vapor, cloud water droplets, liquid water, ice, snow, and graupel) was used. Longwave
and shortwave radiative transfer equations are solved for independent air columns (Fouquart
and Bonnel, 1980; Morcrette, 1991).

40      *P6, line 8: "exchange is controlled"?*
Rewritten:

Atmosphere-surface exchanges are taken into account through physical parametrizations. . .

45      *P6, lines 11-14: The end of the chapter should be improved. For example, for the 1-km and*
*250m domains it is better to say something like "the resolution is high enough for the deep/shallow*
*convection to be solved explicitly" Also, the reference to Table 1 is missing the number and the*

*brackets are left open. The list is missing "and" from the end.*
The sentence that describes domains and used schemes was rewritten:

> Deep and shallow convection parametrization schemes were activated in the coarser domain A. 1-km and 250-m resolutions of domains B and C are high enough for the deep/shallow convection to be represented explicitly.

Number for the reference to the table was added, and mistakes were corrected.

*P6, lines 15-16: remove "files" and ending should be improved (e.g. "for lateral boundary forcing with an update frequency of 6-hours").*
Moved to another paragraph and rewritten:

> European Centre for Medium-Range Weather Forecast (ECMWF) operational analyses, updated every six hours, were used for Meso-NH initialization and domain A boundary forcing

*P6, line 17: "the one" to "one"*
Corrected.

*P6, line 19: remove "-"*
Removed.

*P6, line 20: "have covered" to "covered".*
Corrected.

*P7, line 5: "the freshwater lake..."*
Corrected.

*P7, line 6: "were" to "was".*
Corrected.

*P7, line 15: A new chapter starts so better to say "The initial parameters used in FLake are".*
There are several sets of parameters that are required, so the beginning of the sentence was rewritten:

> FLake model requires at least the following sets of variables and parameters to run...

*P7, line 29: rewrite, a suggestion "the depth of the artificial lakes varies spatially, because".*
Rewritten:

> The depth of the artificial lakes varies decreases rapidly from the center to the shore, because the bottom of the reservoirs used to be valleys.

*P8, lines 8: "to compare" to "analyzed".*
Changed.

*P8, line 10: "of" to "took place between" (and remove second "took place").*

Rewritten:
    The ALEX2014 IOP took place between $22 - 24$ of July 2014 at the Alqueva reservoir.

*P8, line 13: merge this chapter with the first one in section 5.1.*
5  Merged.

*P8, line 14: comma after "point".*
Added.

10  *P8, line 17: rewrite the end of the sentence, e.g. "22 km height; thus, to build a corresponding profile, three..."*
Rewritten:

    Radiosondes reached the altitude of the top of the model (about 22 km) in about 2.5
15  hours.  Therefore in order to build the simulated profile, three consecutive hourly outputs from the model were used.

*P9, line 6: "95.5 %" to "95.5%"*
Corrected.

*P9, line 17: "11.26 %" to "11.26%"*
Corrected.

*P9, line 14: "magnitude are" to "magnitude is"*
25  Corrected.

*P9, lines 11 and 15: "suplementary" to "supplementary"*
Corrected.

30  *P9, lines 15 onwards: make a real list of the statistical values ("following: temperature average bias. . ., humidity average. . ., and for the wind speeds...")*
The sentence was rewritten:

    Statistical results for them are the following: temperature average bias is -0.13 °C, RMSE
35  is 1.49 °C, and correlation coefficient is 0.99; relative humidity average bias is 0.59%, RMSE is 11.26%, and correlation coefficient is 0.87; and for the wind speed average bias is 0.05 m/s, RMSE is 2.07 m/s, and correlation coefficient is 0.90.

*P10, line 1: "accordance" to "accord"*
40  Corrected.

*P10, line 6: "are" is missing. Also, you could add "It should be mentioned that not all...".*
Added both.

45  *P10, line 8: "visible in" should be change to e.g. "which can be seen from".*
Changed.

*P10, lines 10-11: this small chapter can be merged with the previous one.*
Accepted, now it is one paragraph.

*P10, line 10: "100 %" to "100%"*
Corrected.

*P12, line 1: This seems to be a new chapter and yet you referrer to "these stations", please correct.*
Rewritten:

Figure 6 shows the time evolution of simulated and observed air temperature and wind speed at Cid Almeida, Barbosa, and Montante sites.

*P12, lines 23-24: Please rephrase this sentence (starting "Measurements").*
The sentence was rewritten:

Wind direction at ALEX stations is represented in Fig. 8. Different behaviour in wind direction between the two station from 21 to 23 of July is clearly seen from measurements data (green dots). In Barbosa station the wind changes from northwest to south regime during daytime while in Cid Almeida this effect is not observed. In the simulations this difference is not so clear, but is still visible during the afternoon on July 22. Barbosa station, located on the northwest shore of the lake, indicates the presence of the lake breeze because its direction is the opposite to the dominant wind. However, at Cid Almeida station on the southeast shore breeze is co-directed with the dominant wind in the area, so, its appearance is difficult to track.

*P15, line 1: Rephrase, e.g. "To study. . . affects the surrounding area, the following. . . were analyzed in this work:"*
The paragraph was rewritten:

To analyse the impact of the Alqueva reservoir on local area the changes of the following atmospheric variables, such as air temperature and potential temperature, relative humidity and water mixing ratio, and vertical and horizontal wind speed, were considered. In this section only B and C domain datasets were used.

*P15, lines 2-5: Is the sentence starting with "Overall, simulation result..." necessary?*
The sentence was removed. Rewritten version of this paragraph can be found above.

*P15, line 6: A comma after "During daytime" and add "the" before water temperature and air temperature.*
Corrected.

*P16, line 1: A new chapter and you refer with "its" to? Lake breeze should be mentioned here.*
The sentence was rewritten:

The lower layers of air are the first to be affected by the presence of water.

*P16, lines 12-15: Please rephrase the sentence starting with "Maximum of the temperature" It*

*is too long and complicated.*
The sentence was split into two and rewritten:

> The highest impact on the air temperature can be observed in the early afternoon (12:00
> − 14:00 UTC). The boundary layer is cooling down and its height decreases from more than
> 2 km above the land outside to values close to 1 km over the lake surface (Fig. 10 (a)).

*P16, line 22: "nighttime" to "night-time" and you could move the part in brackets to be before
the comma.*
Corrected:

> During the night, the large-scale circulation (Fig. 9 (a), (b)), driven...

*P16, line 31: A new chapter starts, where does "this" referrer to?*
Corrected:

> The breeze intensifies during the afternoon...

*P18, line 3: use "the cross-sections"*
Rewritten:

> Cross-sections S1 and S2 presented in Fig. 12 show...

*P18, line 9 and Fig. 14: "g/Kg" to "g/kg".*
Corrected.

*P19, line 3: "increase" to "increases".*
Corrected.

*P20, line 1: "this zones" to "These zones".*
Corrected.

*P20, line 5: "proeminent" to "prominent".*
Corrected.

*P20, Figure 13: "KG/KG" to "kg/kg"*
Corrected (see *Comment 20*).

*P21, line 5: "figure out" to "resolve".*
Changed.

**References**

[revised manuscript text omitted]

---

## Author Comment (AC2) · 28 Jul 2018

**Breeze effects at a large artificial lake: summer case study**

Authors' responses to the Reviewer 2 comments

Maksim Iakunin[1], Rui Salgado[1], Miguel Potes[1]
*miakunin@uevora.pt*

[1]Department of Physics, ICT, Institute of Earth Sciences, University of Évora, 7000 Évora, Portugal

**Contents**

**Introduction. Document structure**

This document contains authors' responses to the comments of the Reviewer. The document structure is the following:

- Reviewer's comments are numbered and given in *italic font*. General, specific, and technical comments come separately.

- Authors' response follows the comment and starts after **"Response:"** with normal font.

- The text from the article itself (if some changes were done, and if it is reasonable to provide it) is typed with `typewriter font` and separated from the response with an extra blank line.

- *Technical comments and mistakes* are not numbered, and authors' response follows immediately.

Reviewed manuscript with all the corrections is given after all responses. It contains the changes and proposals of **two** Reviewers and was prepared using LaTeXdiff package for better understanding of what was added or removed.

**Anonymous Referee 2**

*General comments*

*This paper investigates changes in atmospheric variables in the area of Lake Alqueva, induced by the filling of this artificial lake in 2004. To identify the changes, two simulations were performed using a mesoscale atmospheric model, the Meso-NH model. In the first experiment, the lake is not present and in the second one, a lake model, Flake, is run in a coupled mode. The authors observed the formation of a lake breeze in the presence of the reservoir and identified impacts on the atmosphere.*

*This study is interesting as it quantifies the effects of a large lake on the weather of the region. The results are nice and innovative, in particular results presented in Fig. 11 and Fig. 13, but I think the author could go a bit further and relate their findings (in terms of simulations) with changes that have been observed at the weather stations. Did they also notice changes in the observed wind regime between 2010-2018 and year1990-2000 for instance? Otherwise, the paper looks more like a first draft, which makes the reading quite painful. Some explanations are too vague, some acronyms are not defined, and many sentences are awkward. I highly recommend that an English speaker reads the manuscript before resubmitting.*

**Response:**  The wind regimes between 2010-2018 and 1990-2000 were not studied. It would be very interesting to make that comparison, but the meteorological stations were installed in the lake shores only during two field campaigns. One in the summer of 2014 (ALEX) and more recently in February 2017 (ALOP — Alentejo Observation and Prediction systems) which is an ongoing experiment.  Since the lake breeze is only detected nearby the lake shores and loses its intensity entering inland we have no chance to make that study because the closer stations operating continuously from our Institute (ICT) and from the Portuguese Institute for Sea and Atmosphere (IPMA) are Portel and Reguengos de Monsaraz (Fig. 1), where the lake breeze effect is not noticeable due to the distance to the lake. In a previous paper, Policarpo et al. (2017)

have studied the effect of the Alqueva reservoir on fog and for this purpose it was possible to use observational data.

Anyway, we try to go a little further in the analysis and introduce data observed in the summer of 2014.

5      The paper was carefully reviewed and explanations were added, acronyms defined, awkward sentences rewritten taking into account a better reading of the manuscript.

**Specific comments**

**Comment 1**

10      *The formation of the lake breeze is not clearly explained.*

**Response:**    Corresponding paragraphs of section **6 Lake impact** were expanded to reflect more information about this effect. Also, some discussion was added to the **7 Conclusions** section.

**Comment 2**

15      Some acronyms are given but they first need to be explained. For instance, in the introduction, you mention NH3D. What king of model is it (ex: atmospheric model)? Same, when you mention Meso-NH model, SURFEX, Flake. As well, p.3: what is Csa? (Mediterranean climate) should appear in the text, and Csa should be in brackets (Csa according to the Köppen climate classification). Again, in p.6 ECOCLIMAP and SRTM. You need to clarify.

20  **Response:**

- NH3D — non-hydrostatic 3-dimensional mesoscale model;

- Meso-NH — non-hydrostatic mesoscale atmospheric model;

- SURFEX — Surface Externalisée, in French;

- SRTM — Shuttle Radar Topography Mission;

25    - ECOCLIMAP — is the name of a database, not an acronym. We could suppose that it came from ECOlogical and CLImate MAP, but the authors do not indicate it;

- FLake model — Freshwater Lake model

Corresponding corrections were made in various parts of the article.

30      `Csa` as well as `Bsk` are categories of climate in Köppen climate classification. The sentence was rewritten:

```
    This region has Mediterranean climate with dry and hot summers (Csa according to the
Köppen climate classification), with a small area within of the mid-latitude steppe (BSk)
35 category.
```

**Comment 3**

*P7. L34: You mention Flake results based on 2-4 months simulations. Did you perform these simulations? Which period did you choose to run these simulations? What is the correlation between simulated and observed data? I would like to see how well the model reproduces the surface temperatures. This is very important to assess the intensity of a lake breeze and the accuracy of the results.*

**Response:** Yes, these simulations were done, and FLake shows very realistic results both in short-term and long-term simulations. Example of the short-term simulations for the IOP was added to Fig. 2:

[Figure]

Corresponding text was added to the section **4.2 FLake model**:

The comparison between measurements of water temperature near surface (at 1 meter depth) and FLake simulated values of mixed layer temperature are shown in Fig. 2 (b). Sensor at 1 meter depth was chosen because it always stays in mixed layer and is not affected by surface "skin" effects. Modelled values are close to measurements which indicates that the initial conditions were realistically imposed.

Long-term simulations was performed for all ALEX data. It was not shown in the article because at the moment, another paper is being prepared on these results and FLake initial parameters. Example for this simulation for 4 months with different FLake shape factors is shown below:

[Figure]

**Comment 4**

*The discussion on the lake effects focuses on the southern part of the lake. Are the conclusions also valid in the northern part of the lake?*

**Response:**   Yes, most of the discussion is centered around the S1 cross-section because it crosses Montante floating platform — the source of measurement data. But S2 cross-section located more to the center of the lake, and the results there are similar. We also studied cross-sections of the middle of the lake in the most wide part (west to east direction) and came to the same results. Breeze effect is observed in the northern part of the lake as well (it is seen on the maps, Fig. 10), but its intensity is not so high due to the fact that the lake there is much more narrow. Maps of the differences of mater vapor mixing ratio provided in the response to the next comment also show that the same conclusions are valid for the central and northern parts of the lake.

**Comment 5**

*You mention that changes in relative humidity are mostly related to change in temperature. However, looking at Fig. 10 and Fig. 12, differences do not appear at the same place. It is maybe related to the fact that the hours on each subplot of figures 10 differ from those on figure 12. It would make sense to have something more homogeneous. Also, wouldn't it be worth adding a map, such as Figs. 10 and 12, representing surface specific humidity? Are they several descending branches of dry air over the lake?*

**Response:**   Following the Reviewer comment, Fig. 12 was replotted with the same time of output as Fig. 10 (note, that one figure was removed from the section **6**, so the numeration was changed).

[Figure]

[Figure]

Night and early morning air temperature anomalies are not high enough to produce significant difference in relative humidity, but now it is easy to catch the relationship between these two variables at daytime.

Also, we add figures with the near surface water mixing ratio and a new paragraph.

Figure 14 illustrates this process in a horizontal plane. At midnight (Fig 14 (b)) the reservoir does not directly affect vapour mixing ratio in the air. In the morning hours, when the sun has risen, but the breeze system has not yet formed, a positive impact on the moisture over the lake can be seen due to the increase of the evaporation. This anomaly affects the air above central and southern part of the reservoir and is advected to other nearby areas (Fig 14 (c)). Later in the afternoon, with the formation of the lake breeze, a negative impact can be traced over the water surface due to the descending branches of the local circulation (Fig. 14 (d, e)). This explains the afternoon decrease of the water vapour mixing ratio observed at the Montante platform as seen in Fig 13. The localization of the area of this negative anomaly

changes in time, but predominantly it is over the larger southern part of the reservoir. With the dissipation of the local lake breeze system and the arriving of the stronger large scale northwestern wind, the negative moisture anomaly over the reservoir disappears and a positive effect is visible on the downwind region (Fig. 14 (a, f)), due to the increase of evaporation (note that Fig. 14 (a) corresponds to the night of July 21 to 22, when the effect was more noticeable).

[Figure]

**Comment 6**

*P.12 You indicate the maximum error in terms of temperature. My feeling is that a bias of 5° is quite a lot and especially when it last for several hours. I would a discussion on the impact of this bias on the turbulent fluxes or some hypothesis in order to explain why the fluxes are so well reproduced considering this bias. This could affect modelled lake surface temperature and the intensity of lake breezes.*

**Response:**   Indeed, maximum error of 5 °C seems to be huge. But in fact, such difference can be observed only during very short time period (at 1 or 2 timesteps, see Fig. 6). More important parameter here that describes the whole period of validation is average bias which value for Cid Almeida station is 0.5 °C (and root mean square error is 1.57 °C). Single relatively big differences,
5    if they occur, do not lead to critical errors in subsequent conclusions.

Latent and sensible heat fluxes were measured at Montante platform, where maximum difference bias was 3.2 °C at late afternoon of July 22. At that time there is a gap in flux measurements, but as it can be seen on Fig. 7, the difference between the model and assumed measured values is the highest. At other intervals of time the biases of air temperature and fluxes are much smaller.
10   Corresponding corrections were added to the article.

**Comment 7**

*The lake effect part is very interesting, but it is hard to follow the mechanism you describe. On Figure 11, you should draw circle where you identify "the upper-level convergent return circulation". The figure needs to be bigger.*

15   **Response:**   Figure 11 (now Figure 10) was enlarged and font size was changed so now it looks better. Wind speed vectors do not merge into one and the circulations are seen much clearer.

[Figure]

**Comment 8**

*In the conclusion, I would expect some general comments on your findings. Are the conditions on July 22-24, representative of the conditions that prevail in this area in summer? What kind of experiments should be done in the future or is there anything you would like to investigate further? What are the limits to your conclusion? There are some biases in the atmospheric variables between modelled and observed data. How confident are you in your results?*

**Response:   Conclusions** section was expanded and more comments were added to the discussion:

This work is dedicated to the studies of the formation and magnitude of the summer lake breeze at the Alqueva reservoir, South Portugal, one of the impacts of the artificial lake on the local weather. The study was based on Meso-NH simulations of a well documented case

study of 22-24 July 2014. This period was taken for several reasons. First, a large volume of meteorological data was collected during these days, which allowed for a validation of the simulation results. Secondly, this period was hot and dry, which is typical for most summer days in this region.

5    The model allowed to conduct the simulation with horizontal resolution of 250 meters which is fine enough to resolve such relatively small scale lake breeze and to spot the impact of the reservoir on the detailed local boundary layer structure. Due to the "youth" of the Alqueva reservoir it is possible to run atmospheric model with the surface conditions prevailing before the filling of the reservoir. Two simulations, one with Alqueva and another one without it,
10    allow to evaluate the raw impact of the lake on the local weather regime.

Formation and dissipation of the daytime breeze system induced by the reservoir are described in the work. On hot summer mornings the difference between air temperatures above water and neighbouring land surfaces induces the radial movement of air from the lake. The breeze system starts to form in the morning and the peak of the wind speed reaches
15    6 m/s in the late afternoon. Simulation results show that the lake breeze could be detected at a distance of more than 6 km away from the the shores and on altitudes up to 300 m above water surface. In late afternoon the dissipation stage of the lake breeze system anticipated with the arrival of the larger scale sea breeze from the Portuguese west Atlantic coast. In early evening ($19{:}00 - 20{:}00$ UTC) the local lake breeze system can not be detected anymore.
20    No reverse land breeze is detected during the night.

During daytime, the simulation testify the observed very low evaporation from water surface ($0 - 120$ Wm$^{-2}$ in terms of sensible heat flux), due to weak winds and the stable stratification of the internal atmospheric surface layer. A night-time, the strong winds associated with the Peninsular larger-scale circulation induced by the sea-land contrasts, induce a very
25    high evaporation rate ($200 - 250$ Wm$^{-2}$).

The cooling effect of the reservoir can decrease the air temperature up to 7 °C, nevertheless is limited by the lake borders and normally can not be seen farther than few kilometers away from the shore mostly in southeast direction. The cooling can be found up to 1200 m above the lake surface.

30    Lake breeze system brings dry air from upper atmospheric layers (2-2.5 km) to near surface levels above the reservoir. This effect leads to the fact that the air above the surface of the lake becomes more dry in terms of water vapor mixing ratio, in spite of its relative humidity can increase up to 50% due to the decrease in air temperature.

Further work implies two directions. The first is tuning the lake model and its initializa-
35    tion in order to obtain more accurate results and reduce validation biases. The second is to carry out a longer experiment, which would cover a 12-month period. Such simulation could reveal seasonal aspects of the impact of Alqueva on local weather.

**Comment 9**

40    *The units are not systematically the same. Temperatures unit are for instance in °C in Fig. 5 but in K in Table 2.*

**Response:**   True, it was corrected, now in the **Table 2** temperature units are °C. Also, the units are °C everywhere if it is referred to air temperature. In the cross-sections, when we discuss potential temperature, the units are Kelvins.

*Comment 10*

P5: you mention 3 domains, A, B, and C, why don't you use these terms later in the text? For instance on P6: Domaine B required deep convection. . . It would make the manuscript easier to read.

**Response:** Thank you for noticing that. Indeed, several times we named domains for their resolution, skipping the notation that we introduced. Now it was corrected throughout the whole article.

*Comment 11*

Some figures are too small. For instance, Figure 5. Also use the same symbol for corresponding stations on each subplot. Figure 11 needs to be bigger.

**Response:** Agree, elements of some figures are unreadable — they were enlarged and corrected. Also now the same symbols are used for corresponding stations in the legend on Fig. 5:

[Figure]

*Comment 12*

Figure 10: you should name the cross section. For instance S1 and S2 and refer to them in the text. That would ease the reading.

**Response:** Yes, this can help to improve the section. Cross-sections named S1 and S2, corresponding references and corrections are made in **6 Lake impact**.

*Comment 13*

Figure 13 and others: it is weird to have different scales for the windspeed. It is then difficult to assess the evolution of the windspeed throughout the day.

**Response:** The figures were replotted with the same reference vector length and value (See *Comment 7* and Figure below).

[Figure]

**Comment 14**

*P8. You say twice that the domain B is used for validation with radiosondes.*

**Response:**   Indeed. The sentence

```
This comparison is done in 1-km horizontal resolution domain
```

was removed.

**Comment 15**

*In the dataset section, try to gather the information per station. Also later in the text (p.11), you define the coordinates of the stations.*

**Response:**  Agreed, coordinates of the stations and the platform should be in the Measurement data section. It was moved there and the first paragraph of the 5.2 section was rewritten:

In addition to the validation against the IPMA synoptic stations, comparisons were made with data obtained at ALEX stations (Barbosa, Cid Almeida and Montante platform). Their coordinates were used to locate corresponding grid points on the C domain output.

**Comment 16**

*P1. Abstract: you say that two simulations have been done with the meso-NH model coupled to Flake. Only one was coupled, no?*

**Response:**  No, the meaning here is the following: version of atmospheric model Meso-NH that we use is coupled to FLake model. It is this combination was used in all simulations.

**Comment 17**

*P1: L 25: daily air temperature near the surface is decreased in lake shore areas -> and above the lake?*

**Response:**  Thank you for pointing this out, the sentence was corrected:

Normally, near surface relative humidity is increased while daily air temperature is decreased above lake and shore areas.

**Comment 18**

*P1. L26: lake surface balances the atmosphere above ->clarify*

**Response:**  The sentence was rewritten:

During the warm summer periods relatively colder lake surface interacts with the atmosphere above, which leads to a reduction of clouds and precipitation.

**Comment 19**

*P2. L3: In autumn and winter it has the opposite effect due to the warmer air above lake surface: increase of evaporation and cloud formation -> not warmer air above the lake in summer?*

**Response:**   Agree, the phrase is incorrect. The sentence was rewritten:

In autumn and winter it has the opposite effect: due to the fact that water is warmer than the air above, increase of evaporation and cloud formation can be observed (Ekhtiari et al., 2017).

**Comment 20**

*P2. L32: Simulation has been done for. . . -> which simulation are your talking about? A simulation performed within the ALEX2014 experiment?*

**Response:**   Indeed, this sentence was unclear. Rewritten:

In this work simulations have been done for the Intensive Observation Period (IOP) of ALEX project (ALqueva hydro-meteorological EXperiment, http://www.alex2014.cge.uevora.pt/).

**Comment 21**

*P4, L24: 3 stations of Instituto de Ciencias etc. . . -> what kind of stations? What kind of variables?*

**Response:**   Weather stations (added to the text).

**Comment 22**

*ALEX and ALEX2014, is it the same database?*

**Response:**   Yes, it is one database from one experiment. To avoid further confusions we changed ALEX2014 to ALEX throughout the text.

**Comment 23**

*P4. The two land stations your refer to, are they the weather stations you mentioned earlier? Gather the information and be consistent. Alqueva Montante and Montante, the same?*

**Response:**   Yes, the main point here is about land weather stations (named Barbosa and Cid Almeida) and floating platform named Montante. Several corrections were made in these paragraphs to make it clearer.

**Comment 24**

*You say that the choice of your study period is based on atmospheric conditions. But you also say that the project lasted for 3 days. Wasn't the choice more based on the availability of data?*

**Response:** Not exactly. ALEX lasted from June to October 2014 and included an Intensive Observation Period (IOP) — these three days (22-24) in July. Atmospheric data from stations and floating platform are available for the whole ALEX period. IOP period was chosen for case study because **a:** this period reflects a typical summer weather in the region, and **b:** additional data from radiosondes was available for validation.

**Comment 25**

*P. 9: the worse values are in the lower lever. What do you mean? Extremely bad?*

**Response:** No, just relatively worse. The paragraph was rewritten to be more clear:

The principal features of the profiles trend are well represented by the model. During daytime, air temperature and relative humidity curves indicate that the model tends to well represent the height of the boundary layer at 2-2.5 km altitude (around 2 km in Fig. 4 (a), (b)). Overall, Meso-NH reproduces the air temperature above the surface layer (over 500 m) very well. Near surface, the Meso-NH tends to anticipate the development of the unstable boundary layer in the morning (9:00 and 12:00 UTC), simulating higher temperatures in the lower levels. In the late afternoon (18:00 and 21:00 UTC) the model also tends to anticipate the decrease of the temperature in the surface layer (see the supplementary material, Fig. S1).

**Comment 26**

*p. 9: patterns look similar. Are they similar or do they just look similar?*

**Response:** In fact, we should admit that modelled curves reflect only principal changes in these variables, e.g. low level jet, and, in general, more smooth than measured curves. That is why we say that, in general, patterns of modeled and measured curves look similar. However profile validation showed good results.

**Comment 27**

*P.16: The first level of air above the lake is the most affected by its impact- > impact of what?*

**Response:** The sentence was rewritten for better understanding:

The lower layers of air are the first to be affected by the presence of water.

**Comment 28**

*P16: Need to clarify where you mention positive or negative anomalies. Over the lake?, over the land surface?.*

**Response:** To avoid a confusion the following sentence was added:

By positive and negative anomalies here we mean the differences between LAKE1 and LAKE0 simulations.

We do not consider it over the land or lake surface, in general it is the area where the difference is not zero.

**Comment 29**

*P.18, legend: what do you mean with projection of wind, same for figure 13.*

**Response:**  By projection of wind we mean the component of the wind vector in the plane of the cross-section.

**Comment 30**

*P.18: the fact RH is decreasing due to change in temperature is an important point. Remove "it should be noted"*

**Response:**  Agree, "it should be noted" was removed from this sentence.

**Technical corrections**

*Sentences that need to be rephrased*

*P.2 the using of coarse spatial resolution observations data*
Rewritten:

. . . inexistence of observational data at sufficiently fine spatial resolution.

*On many pages: meteorological variables instead of parameters*
Agree, in many places we abused the word "parameters". We replaced it with the "variable" in the proper places.

*p.2, L20. Remove "in his PhD thesis also in Portuguese", not relevant*
Removed. Updated sentence:

The studies were continued and improved by Salgado (2006) who did the first attempt to quantify the direct effect of the reservoir on the local climate, in particular on winter fog.

*p2, L 27. Surface models Masson et al used among atmospheric models by Meso-NH*
This paragraph was removed from the article.

*p2. L 29: allows to gain the results*
The sentence was rewritten:

Mesoscale atmospheric models, such as Meso-NH, allow to obtain results with sufficient horizontal resolution (250 m in present study) for studying the local effects of air temperature changes and the generation of small-scale circulations under different large-scale atmospheric situations.

*Many times, you use ":", make a sentence that includes what follows. For instance, you could replace the ":" by "such as "*
Yes, colons were used way too many times. In the places where it was not necessary it was removed from text or replaced with "such as" construction.

*P4: L3: "on the surface level? −> at the surface*
Rewritten:

The incident solar radiation at the surface is...

*P5: L10: For that, ->remove.*
Removed. Two sentences were merged into one:

The intensive observation period of the ALEX project lasted 3 days (22-24 July) and included launches of meteorological balloons every 3 hours.

*P. 6:Longwave and shortwave radiative transfer equations are solved for independent air columns*
Rewritten:

Longwave and shortwave radiative transfer equations are solved for independent air columns...

*P6: A set of two numerical simulations were performed. . .*
Corrected:

To track the impact of the reservoir on the weather conditions two numerical simulation were performed...

*P.9, L15: Temperature average bias is -0.13 K, RMSE 1.49 K, and correlation coefficient is 0.99. Humidity average bias is 0.59% RMSE of 11.26 % and 0.87 correlation coefficient.*
The sentence was rewritten:

Statistical results for them are the following: temperature average bias is -0.13 °C, RMSE is 1.49 °C, and correlation coefficient is 0.99; relative humidity average bias is 0.59%, RMSE is 11.26%, and correlation coefficient is 0.87; and for the wind speed average bias is 0.05 m/s, RMSE is 2.07 m/s, and correlation coefficient is 0.90.

*P10. Scatter plots of air temperature, relative humidity, and wind speed shown on Fig. 5 -> verb missing*
Corrected:

Scatter plots of air temperature, relative humidity, and wind speed are shown in Fig. 5

*P.10: The worse result are observed in comparison against Portalegre data*
The sentence was rewritten:

5      The worst results are observed in Portalegre data...

*P.11; Legend needs to be clarified. Comparison of modelled air temperature. . .with*
The legend was rewritten:

10     Scatter plots of the comparison between Meso-NH sumulation LAKE1 and measured values at synoptic stations. Air temperature (a), relative humidity (b), and horizontal wind speed (c).

*p.11. L 7-8, suggestion: In the case the meteorological stations were located in a lake grid cell,*
15 *the nearest land . . .*
Thank you for this suggestion. The sentence was rewritten:

In the case of land stations with grid point associated to water fraction, the nearest land grid point was chosen.

*P12. L2-3: Meso-NH underestimation of air temperature in the afternoon time is opposite of wind speed overestimation at the same period.*
The part was rewritten:

25     Overall, the simulation results are slightly more conservative (except wind speed over the Montante platform), but in general, the patterns are well represented. The model could not represent well the maximum and minimum temperatures, especially in land stations where the temperature range is larger. Regarding wind speed, the model underestimates the maximum values at land stations and at Montante platform (Fig. 6 (d)), on the contrary, the wind
30 speed is overestimated by the model, but the principal features of the curve is represented.

*P15, L3: is a 3 sets of -> consist of*
There was no need in this sentence in the article, and it was removed. The rest of the paragraph was rewritten:

To analyse the impact of the Alqueva reservoir on local area the changes of the following atmospheric variables, such as air temperature and potential temperature, relative humidity and water mixing ratio, and vertical and horizontal wind speed, were considered. In this section only B and C domain datasets were used.

*P15, L3: (for each horizontal resolution) > (one set per domain)*
See the comment above.

*P16, L14: which depth decrease (very clear seen on Fig.11). I don't know what you mean.*
45 This part was rewritten:

The highest impact on the air temperature can be observed in the early afternoon (12:00

– 14:00 UTC). The boundary layer is cooling down and its height decreases from more than 2 km above the land outside to values close to 1 km over the lake surface (Fig. 10 (a)).

*Typing errors and other mistakes*

5      *p2. L20: a first attempt were done*
The sentence was rewritten:

      . . . who did the first attempt to quantify. . .
      *p3. L7: "if" instead of "of"*
10  Corrected.

      *p3, L8: 92 m instead of 92.0 m*
Corrected.

15      *p4. L31: locaton*
Corrected.

      *p6. Table: convction*
Corrected.

      *p7: Flake were used*
Corrected.

      *p9, L5: accurace*
25  Corrected.

      *p9. L11 et L15: supplementary*
Corrected.

30      *p.9 Statistical results are following*
Corrected:

      Statistical results for them are the following: . . .

35      *p. 11: meteostation*
Replaced by weather stations.

      *p. 12, L19: minimums*
Corrected.

      *p.12, L22: are tend to*
Corrected.
      *p.20, L1: this zone -> This zone*
The sentence rewritten:

On the other hand, Fig. 12 also shows that outside the reservoir there are zones of low-level convergence and upward motion that increase the moisture of the boundary layer and form some kind of lake breeze fronts.

*p.20, L4: midle*
Corrected: `middle`

*P.10 L14: lesser*
Corrected: `less`

*P.10 L13. Verb missing*
The end of the sentence changed to:

`... with bias always less than 1 degree.`

*P.16, L29: the teservoir*
Corrected.

*P.16, L31: intensifes*
Corrected.

*P.21: more wet*
This paragraph was rewritten:

[revised manuscript text omitted]

---

## Author Response (AR2)

**Breeze effects at a large artificial lake: summer case study**

Authors' responses to the additional Reviewer 1 comments

Maksim Iakunin[1], Rui Salgado[1], Miguel Potes[1]
*miakunin@uevora.pt*

[1]Department of Physics, ICT, Institute of Earth Sciences, University of Évora, 7000 Évora, Portugal

**Contents**

**1    Introduction. Document structure**

This document contains authors' responses to the comments of the Reviewer. The document structure is the following:

- Reviewer's comments are numbered and given in *italic font*. General, specific, and technical comments come separately.

- Authors' response follows the comment and starts after **"Response:"** with normal font.

- The text from the article itself (if some changes were done, and if it is reasonable to provide it) is typed with `typewriter font` and separated from the response with an extra blank line.

**3    Anonymous Referee 1**

**1        General comments**

*The manuscript has improved from the first phase and is nearly ready for final publishing. The authors have answered successfully to all of the comments I made in the first phase and done the requested modifications to the manuscript. There are, however, few minor issues to be solved before proceeding further.*

**2        Technical comments**

**2.1    Comment 1**

*The language has improved but there are still some places where the text is hard to follow. I again suggest that the authors get editing help from someone with full professional proficiency in English.*

**Response:** manuscript was re-reviewed one more time by all authors and several minor corrections regarding the language.

**2.2    Comment 2**

*You seem to cite the supplementary Fig. S3 before S2 (so please change the numbering). Also, I cannot find any citation to supplementary figures S4-S6 from the text.*

**Response:** Numbering for S2 and S3 was changed (S2<−>S3). References to supplementary figures S4-S6 were put into the text at corresponding places.

**2.3    Comment 3**

*You named (as the referee 2 asked) the cross sections to S1 and S2. This is a bit confusing when you also cite supplementary figures S1 and S2. I would change the references to the cross sections to something else (I recommended roman numbers I and II in the first phase). Actually, it would be nice to see the names of the cross sections in the figure as well (next to the lines).*

**Response:** Cross-sections were renamed to Cs1 and Cs2 to avoid confusing and corresponding labels added to Fig. 9:

[Figure]

**2.4   Comment 4**

*You made the interpolation of wind speeds from 10 m height to 2 m. I think this should be mentioned in the text and the method should be cited.*

**Response:** Used approach is well-known and we decided that no citation needed. The following text was added:

Regarding wind speed, Meso-NH provides values for 10 meters height while the measurements were done at 2 meters. For proper comparison modelled values were interpolated to the height of the sensors using known logarithmic approach and a roughness length (which was also got from the model).